# Rarity is a more reliable indicator of land-use impacts on soil invertebrate communities than other diversity metrics

Andrew Dopheide[1]*, Andreas Makiola[2], Kate H Orwin[3], Robert J Holdaway[3], Jamie R Wood[3], Ian A Dickie[4]

[1]Manaaki Whenua - Landcare Research, Auckland, New Zealand; [2]Bio-Protection Research Centre, Lincoln University, Lincoln, New Zealand; [3]Manaaki Whenua – Landcare Research, Lincoln, New Zealand; [4]Bio-Protection Research Centre, School of Biological Sciences, University of Canterbury, Christchurch, New Zealand

**Abstract** The effects of land use on soil invertebrates – an important ecosystem component – are poorly understood. We investigated land-use impacts on a comprehensive range of soil invertebrates across New Zealand, measured using DNA metabarcoding and six biodiversity metrics. Rarity and phylogenetic rarity – direct measures of the number of species or the portion of a phylogeny unique to a site – showed stronger, more consistent responses across taxa to land use than widely used metrics of species richness, effective species numbers, and phylogenetic diversity. Overall, phylogenetic rarity explained the highest proportion of land use-related variance. Rarity declined from natural forest to planted forest, grassland, and perennial cropland for most soil invertebrate taxa, demonstrating pervasive impacts of agricultural land use on soil invertebrate communities. Commonly used diversity metrics may underestimate the impacts of land use on soil invertebrates, whereas rarity provides clearer and more consistent evidence of these impacts.

*For correspondence:
dopheidea@landcareresearch.co.nz

Competing interests: The authors declare that no competing interests exist.

## Introduction

Land-use changes through deforestation, agricultural development, and urbanisation have caused worldwide impacts on the biodiversity of terrestrial communities and ecosystems (*Dirzo et al., 2014*; *Newbold et al., 2015*). Invertebrates are the most diverse and abundant component of animal biodiversity worldwide and are major contributors of terrestrial ecosystem services such as pollination, soil formation, and nutrient cycling (*Lavelle et al., 2006*; *Wagg et al., 2014*; *Yang and Gratton, 2014*). Long-term declines in the richness and biomass of insects and other terrestrial invertebrates are predicted to have major impacts on food webs and ecosystem functions (*Eisenhauer et al., 2019*; *Hallmann et al., 2017*; *Potts et al., 2010*). Despite this, most invertebrate species remain undescribed, and there is an incomplete understanding of land-use effects on invertebrate biodiversity, particularly for those that reside in soils (*Cameron et al., 2018*; *Eisenhauer et al., 2019*).

Biodiversity loss is typically measured as reductions in species richness (i.e., total number of species; e.g. *Forister et al., 2010*; *George et al., 2019*; *Newbold et al., 2015*). Despite widespread concern about biodiversity loss, evidence for impacts of anthropogenic land use on terrestrial invertebrate species richness is mixed, with studies often detecting richness declines for some taxa or groups but not others (*Allan et al., 2014*; *Attwood et al., 2008*; *Blaum et al., 2009*). Among the few studies that have examined land-use impacts on below-ground invertebrate communities, one detected negative impacts of long-term disturbance on soil invertebrate richness (*Callaham et al., 2006*), another detected increasing alpha diversity and homogenisation of soil invertebrates with increasing grassland intensification (*Gossner et al., 2016*); while others detected inconsistent richness patterns among different soil invertebrate taxa across land uses (*George et al., 2019*;

**eLife digest** Living within the Earth's soil are millions of insects, worms and other invertebrates, which help keep the ground healthy and fertile. There is a growing concern that changing land-use habits, such as agriculture and urban development, are causing these populations of invertebrates to decline. However, to what extent different types of land use negatively impact soil invertebrates is not clear.

Healthy habitats often have a greater variety of species. This biodiversity can be measured in a number of ways, ranging from counting the number of species, to more complex approaches that calculate a species' role in an ecosystem or how close it is to extinction. Finding a way to sensitively measure the biodiversity of soil invertebrates could further researcher's understanding of how different types of land use are affecting these communities.

A new method known as DNA metabarcoding has made it easier to distinguish between different species and calculate the biodiversity of entire populations. Now, Dopheide et al. have used this technique to study invertebrate communities from 75 sites across New Zealand which have been impacted by different land-use habits. This revealed that the most reliable and consistent way to uncover how land use affects soil invertebrates was to measure the rarity of species (i.e. the number of unique species present at each site).

Dopheide et al. found that agriculture negatively affected soil invertebrates and that most types of invertebrates responded in a similar way. Horticulture – such as orchards and vineyards – had the most severe impact, with the lowest variety of species compared to grassland or forest.

Other measurements of biodiversity, such as the number of different species, may underestimate the negative impact agriculture is having on invertebrate communities. The findings of Dopheide et al. highlight why developing strategies to preserve and restore these communities is so important. However, more work is needed to understand what specifically is causing biodiversity to decline and how this effect can be reversed.

*Tsiafouli et al., 2015*; *Wood et al., 2017*). These inconsistent patterns make it difficult to draw general conclusions about the impacts of land use on soil invertebrate biodiversity (*Allan et al., 2014*), and make the use of individual taxa as bioindicators problematic (*Gerlach et al., 2013*).

Inconsistent patterns in biodiversity measurement may reflect limitations of the diversity index used. In particular, species richness provides no indication of the distribution, taxonomy or function of species or communities (*Fleishman et al., 2006*; *Hillebrand et al., 2018*), potentially overlooking the nature and extent of land-use impacts on soil invertebrate communities. In contrast, rarity (sometimes termed 'endemism richness'; *Kier et al., 2009*) measures the extent to which species are widely distributed generalists or limited to particular sites or land-use types. Rarity may thus indicate homogenising effects of land use on communities (*McKinney and Lockwood, 1999*; *Smart et al., 2006*), and the conservation value of sites (*Kier and Barthlott, 2001*). Furthermore, rare species can contribute disproportionately to ecosystem functioning (*Dee et al., 2019*; *Leitão et al., 2016*; *Lyons et al., 2005*; *Mouillot et al., 2013*). Rarity may therefore more accurately reflect the impacts of land use on soil invertebrate communities than species richness.

Rarity and other diversity metrics can also be placed in a phylogenetic context. Phylogenetic diversity reflects the evolutionary history and taxonomic range of communities and associated traits and functions (*Faith, 1992*), thus providing robust information for conservation assessment purposes (*Faith, 1992*; *Forest et al., 2007*; *González-Orozco et al., 2015*; *Mishler et al., 2014*). Phylogenetic diversity can also act as a proxy for functional diversity, albeit imperfectly (*Mazel et al., 2018*; *Srivastava et al., 2012*; *Winter et al., 2013*). Phylogenetic rarity, calculated as the portion of a phylogeny that is unique to a region or habitat (*Mishler et al., 2014*; *Rosauer et al., 2009*), combines elements of both rarity and phylogenetic diversity; high phylogenetic rarity implies that a community contains a taxonomically distinct assemblage of species and associated ecosystem functions. Mean pairwise distance, meanwhile, measures the phylogenetic relatedness of species within a community, which may reflect land-use driven filtering or competitive exclusion processes (*Webb et al., 2002*). The additional information represented by rarity and phylogenetic biodiversity metrics suggests that land-use related patterns based on these values may be clearer and more consistent among soil

invertebrate taxa than those based on species richness and other non-phylogenetic diversity measures. Furthermore, rarity and phylogenetic rarity may be more sensitive indicators of land-use impacts on soil invertebrate communities than richness or phylogenetic diversity, because the former metrics reflect the distribution of species and lineages whereas the latter do not. These possibilities remain untested.

Here we present a comprehensive analysis of soil invertebrate biodiversity across different land-use types at a national spatial scale. We use modern DNA metabarcoding methods to measure invertebrate responses, as this enables the rapid and detailed identification of large numbers of invertebrate specimens from multiple taxonomic groups simultaneously (*Drummond et al., 2015*; *George et al., 2019*; *Wood et al., 2017*; *Yu et al., 2012*) and allows more efficient calculation of biodiversity metrics than previously possible. We analysed the invertebrate faunas in soil samples collected from 75 sites distributed across five different major land-use categories (natural forest, planted forest, low-producing and high-producing grassland, and perennial cropland) throughout New Zealand. Based on these data, we calculated six different biodiversity metrics: species richness, effective species numbers, rarity, phylogenetic diversity, phylogenetic rarity, and mean pairwise distance; as well as standardised effect size (SES) values for the latter phylogenetic metrics. We used these metrics to assess the impacts of land use on a comprehensive range of soil invertebrate taxa. We tested the following hypotheses: 1) all soil invertebrate taxa show the same biodiversity trends across the five land-use types; 2) patterns of soil invertebrate rarity, phylogenetic diversity, and phylogenetic rarity across the five land-use types are more consistent among taxa than species richness or non-phylogenetic diversity; 3) rarity and phylogenetic rarity of soil invertebrates are more sensitive to land use than richness, diversity, or phylogenetic diversity.

## Results

### Overall community composition

We detected a total of 11,284 operational taxonomic units (OTUs), of which 4549 (40.3%) were identified as terrestrial invertebrates. The remainder were identified as protists (37.6%), fungi (14.9%), non-terrestrial metazoans (5%), bacteria (1.7%), and plants (0.5%). The terrestrial invertebrate OTUs mostly belonged to the phylum Arthropoda (2,626 OTUs, among which insects were most common), followed by Rotifera (772 OTUs), Nematoda (656 OTUs), Mollusca (219 OTUs), Annelida (204 OTUs), Platyhelminthes (44 OTUs), Tardigrada (22 OTUs), Gastrotricha (four OTUs), and Onychophora (two OTUs) (*Appendix 1—figures 1* and *2*).

Non-metric MDS ordinations showed clear differences between overall invertebrate community composition in samples from different land-use categories (*Figure 1*). Natural forest samples formed a distinct cluster with no overlap with any other land-use categories. Samples from the other four land-use categories overlapped, with planted forest communities most similar to those from low-producing grassland followed by high-producing grassland communities, and least similar to those from perennial cropland. Similar trends were observed when only Arthropoda, Mollusca, Nematoda, or Rotifera OTUs were included, whereas Annelida OTUs showed less distinction between land-use categories. PERMANOVA tests for composition differences among different land-use categories detected a significant difference based on the overall invertebrate community ($F_{4,61}$ = 1.804, p≤0.001), and based on each of the main phyla detected (Annelida, Arthropoda, Mollusca, Nematoda and Rotifera; $F_{4,44-61}$ = 1.447–2.288, p≤0.001; *Figure 1—source data 1A*).

To test for homogenisation effects of land use on soil invertebrate communities we compared multivariate heterogeneity/homogeneity of sample dispersions, mean pairwise beta diversity, and mean pairwise phylogenetic beta diversity, between land-use categories. For overall invertebrate communities, each of these measures differed significantly among land uses ($F_{4, 61-442}$ = 3.59–14.99, p≤0.011), being highest in natural forest sites and lowest in grassland and/or cropland sites (*Figure 1—source data 1B*; *Figure 1—figure supplements 1–3*). Similar trends were observed for Arthropoda and Nematoda communities based on all three measures, and for Annelida and Mollusca communities based on phylogenetic beta diversity and multivariate heterogeneity of sample dispersions, whereas Rotifera communities showed different patterns.

A heatmap based on the 1000 most relatively abundant terrestrial invertebrate OTUs detected suggested that low-producing grassland, high-producing grassland, and perennial cropland samples

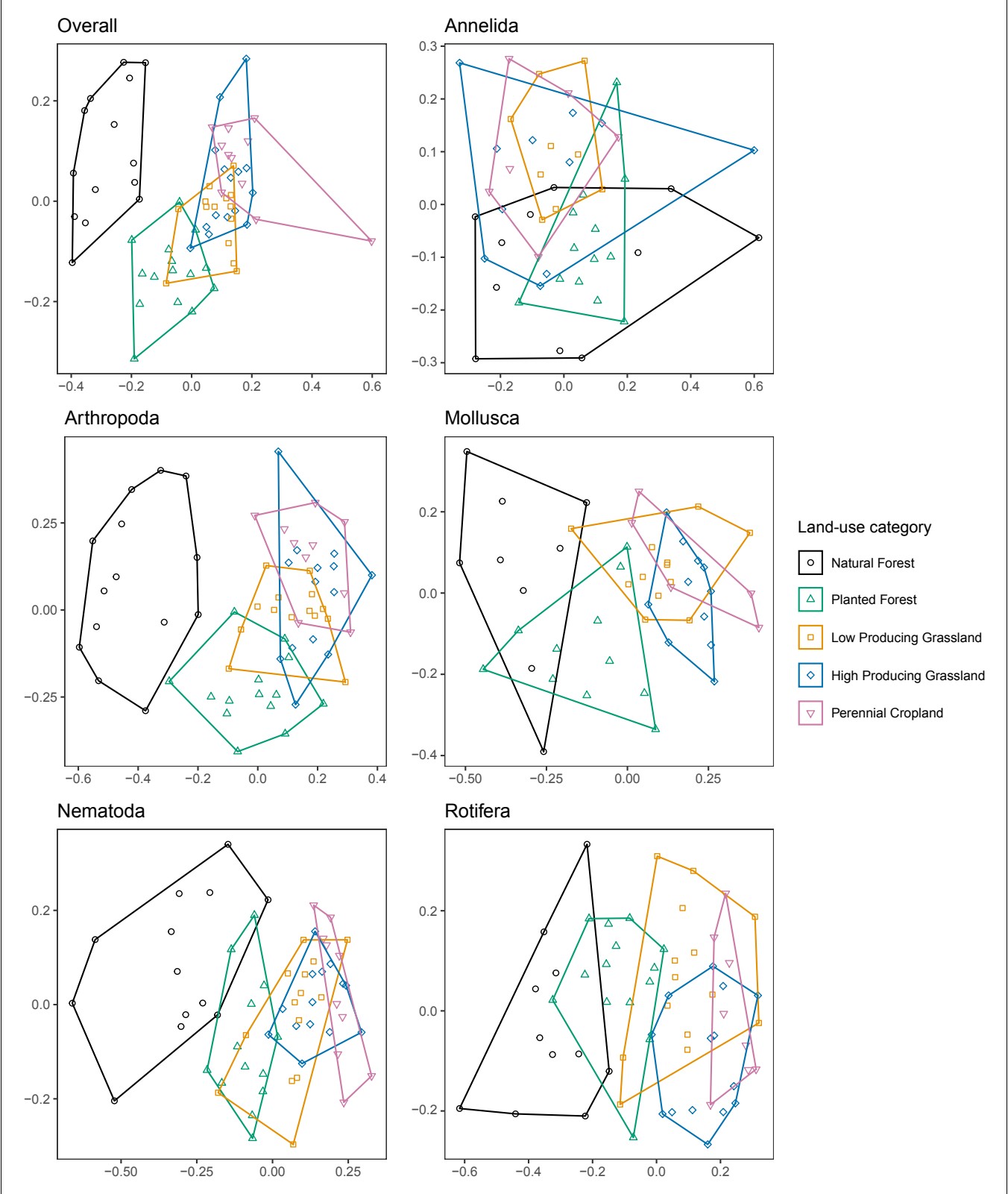

**Figure 1.** Soil invertebrate community composition differs between land-use categories. Non-metric MDS ordinations showing differences in the composition of soil invertebrate communities detected by DNA metabarcoding in five land-use categories, for overall communities, and for individual phyla with ≥ 100 OTUs. Ordinations are based on binary Jaccard distances.

*Figure 1 continued on next page*

*Figure 1 continued*

The online version of this article includes the following source data and figure supplement(s) for figure 1:

**Source data 1.** Results of PERMANOVA tests for differing soil invertebrate community composition, and ANOVA tests for differing multivariate homogeneity of sample dispersions, beta diversity, and phylogenetic beta diversity, between five land-use categories.
**Figure supplement 1.** Multivariate homogeneity of soil invertebrate communities detected in different land-use categories.
**Figure supplement 2.** Beta diversity of soil invertebrate communities detected in different land-use categories.
**Figure supplement 3.** Phylogenetic beta diversity of soil invertebrate communities detected in different land-use categories.

each had relatively consistent assemblages of abundant OTUs, both within and between each land-use category, whereas planted forest samples, and especially natural forest samples, each had more distinctive assemblages of abundant OTUs (*Figure 2* and *Figure 2—figure supplement 1*). In particular, most of the natural forest samples had a subset of abundant OTUs that were not detected in any other sample.

## Overall invertebrate biodiversity differences among land-use categories

All biodiversity metrics (except for mean pairwise distance) showed a general trend of declining overall invertebrate biodiversity (i.e. the biodiversity of the entire invertebrate community) from forested and/or low-producing grassland sites to high-producing grassland and/or perennial cropland sites (*Figures 3* and *4*). Rarity and phylogenetic rarity metrics showed the largest and most consistent land-use-related biodiversity declines, with the highest mean values in natural forest sites followed by planted forest sites and low-producing grassland sites, and high-producing grassland sites, and lowest values in perennial cropland sites. Removing species found in only a single site did not substantially change these trends (*Appendix 1—figures 3–5*). Significant differences between mean biodiversity of overall invertebrate communities in different land-use categories were detected according to richness, rarity, phylogenetic diversity, phylogenetic rarity, and phylogenetic diversity and rarity SES metrics ($F_{4,64}$ = 3.56 to 17.986, p = 0.012 to <0.001), but not effective species numbers, mean pairwise distance, or mean pairwise distance SES metrics (*Figure 3—source data 1A*; *Figure 4—source data 1A*). ANOVA tests of derived land-use rank trends provided similar results, with significant trends identified for all metrics except for mean pairwise distance and mean pairwise distance SES ($F_{1,67}$ = 4.66–31.94, p = 0.034 to <0.001; *Appendix 1—table 1*).

The mean rarity of overall invertebrate communities was significantly lower in all four other land uses compared with natural forest ($t_{23-27}$ = −31.6 to −62.4, *P.adj* = 0.03 to <0.001). Similarly, the mean phylogenetic rarity of overall invertebrate communities was significantly lower in all four other land-use categories compared with natural forest ($t_{23-27}$ = −3.34 to −6.90, *P.adj* = 0.043 to <0.001), and in perennial cropland compared with planted forest ($t_{24}$ = −3.55, *P.adj* = 0.046). In contrast, the mean richness and phylogenetic diversity of overall invertebrate communities were similar in natural forest, planted forest, and low-producing grassland samples, and significantly lower in perennial cropland compared with natural forest ($t_{23}$ = −78.3, *P.adj* = 0.023, and $t_{23}$ = −14.6, *P.adj* = 0.008, respectively) and compared with low-producing grassland ($t_{23}$ = −84.2, *P.adj* = 0.012, and $t_{23}$ = −13.3, *P.adj* = 0.019, respectively; *Figure 3*). Mean phylogenetic diversity SES was significantly lower in low-producing grassland compared with natural forest ($t_{23-27}$ = −2.20, *P.adj* = 0.048), but did not otherwise differ between land-use categories, while phylogenetic rarity SES differences between land-use categories matched those based on non-SES phylogenetic rarity ($t_{23-27}$ = −3.68 to −8.61, *P.adj* = 0.031 to <0.001; *Figure 4*).

A mixed-model ANOVA test for effects of derived land-use rank, land-use category, and taxonomic group effects showed that derived land-use rank and taxonomic group (and interactions) were the most consistently significant predictors of the diversity metrics ($F_{1-16}$ = 7.74 to 32.14, p = 0.007 to <0.001; *Appendix 1—table 2*). The further addition of land-use category to models already containing derived land-use rank did not explain additional variation for effective species, rarity, phylogenetic rarity and mean pairwise distance, but did for richness and phylogenetic diversity (in the form of significant interactions between land-use category and taxonomic group; $F_{48}$ = 1.41 and 1.82, p = 0.037 and <0.001).

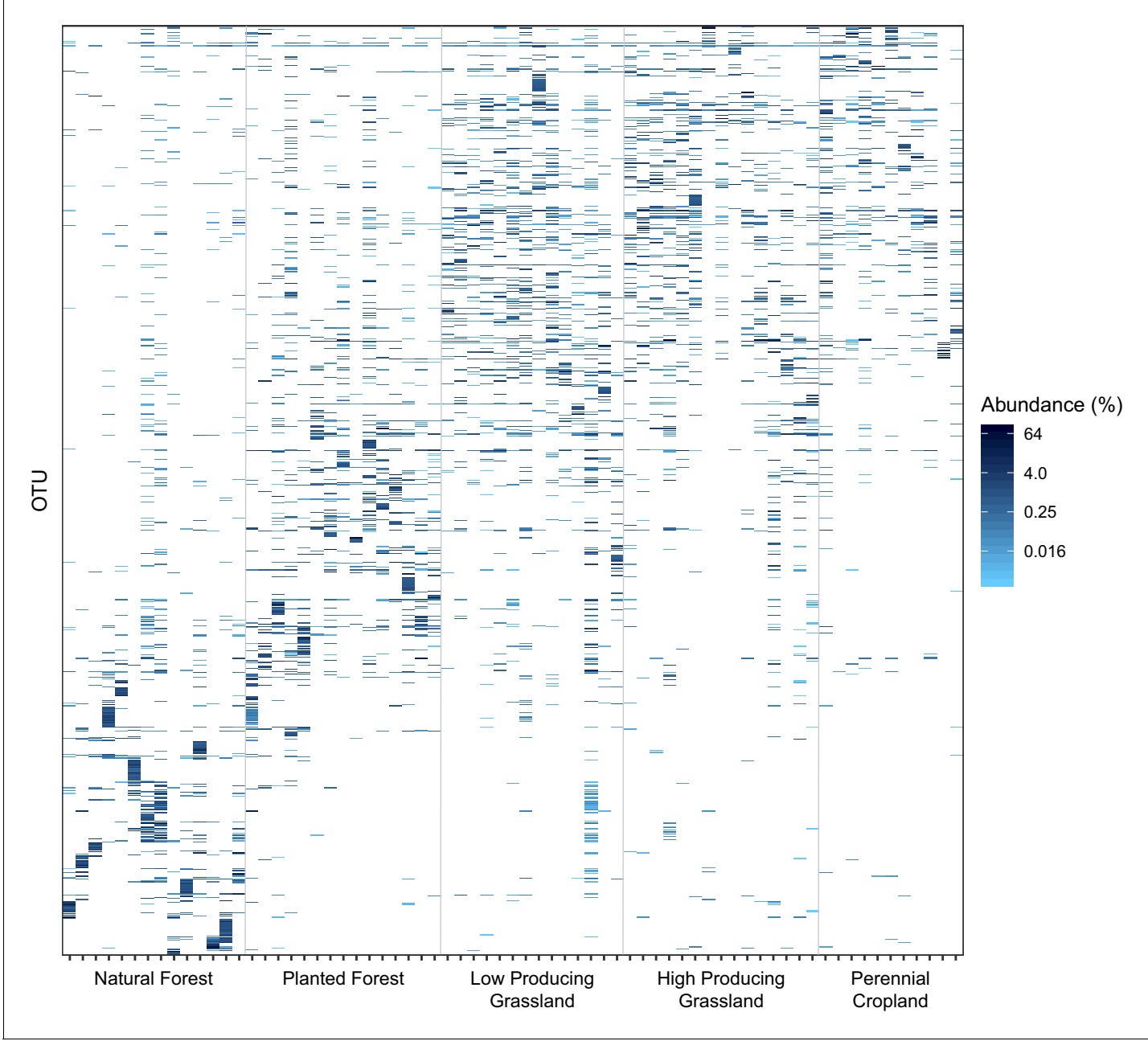

**Figure 2.** Distribution of the 1000 most abundant soil invertebrate OTUs across samples and land-use categories. The proportional abundance and distribution among samples and five land-use categories of the 1000 most proportionally abundant soil invertebrate OTUs detected by DNA metabarcoding, showing that natural forest sites have more heterogeneous assemblages of soil invertebrate OTUs than agricultural sites. Samples are ordered on the x-axis by land-use category and increasing latitude.

The online version of this article includes the following figure supplement(s) for figure 2:

**Figure supplement 1.** Distribution of the 1000 most abundant soil invertebrate OTUs across samples and land-use categories, with samples ordered by compositional similarity.

Most environmental variables showed clear land use-related trends of increasing or decreasing values in the order of natural forest, planted forest, low-producing grassland, high-producing grassland, and perennial cropland (*Appendix 1—figure 6*). An ANOVA test of spatial attributes (latitude and altitude) plus land-use category showed latitude had no effect on overall soil invertebrate biodiversity according to any metric, whereas altitude had significant effects on biodiversity of all metrics

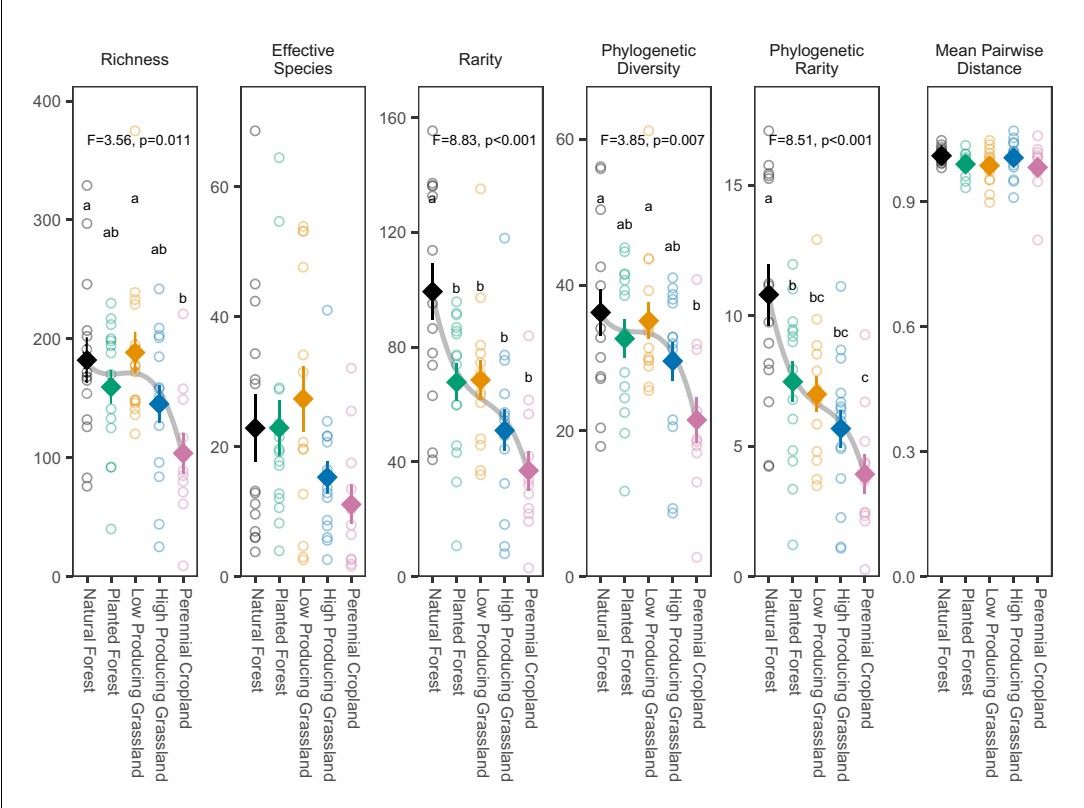

**Figure 3.** Biodiversity estimates for overall soil invertebrate communities detected in different land-use categories. The biodiversity of soil invertebrate communities detected by DNA metabarcoding declines from forested to agricultural sites according to most metrics, with the clearest declines shown by rarity metrics. Diamonds and whiskers represent mean values ± standard errors, with individual data points represented by circles. ANOVA test statistics and trend splines are shown for cases with statistically significant biodiversity differences among land-use categories, with letters indicating differences between land-use categories detected by post-hoc Tukey HSD tests.

The online version of this article includes the following source data and figure supplement(s) for figure 3:

**Source data 1.** Results of ANOVA tests for differing soil invertebrate biodiversity between different land-use categories, according to six biodiversity metrics.

**Figure supplement 1.** Biodiversity estimates for soil arthropod groups in different land-use categories.

**Figure supplement 2.** Biodiversity estimates for non-arthropod soil invertebrate phyla in different land-use categories.

except for mean pairwise distance ($F_1$ = 9.41 to 22.33, p = 0.003 to <0.001). In addition to altitude, land-use category had a significant effect only on rarity and phylogenetic rarity metrics ($F_1$ = 4.40 and 4.60, p = 0.003 and 002; *Appendix 1—table 3*). The first three components of a PCA incorporating latitude, altitude, and soil chemistry variables explained 70.25% of variance. According to an ANOVA test of these three PCA components plus land-use category, the first component had significant effects on the rarity, phylogenetic diversity and phylogenetic rarity of the overall soil invertebrate biodiversity ($F_1$ = 4.79 to 15.25, p = 0.032 to <0.001), and the second component on the former three metrics plus richness ($F_1$ = 7.00 to 10.24, p = 0.010 to 0.002). The third component did not have a significant effect on any of the metrics. The addition of land-use category to these models explained further variation for richness, rarity, and phylogenetic rarity metrics only ($F_4$ = 2.71 to 4.72, p = 0.038 to 0.006; *Appendix 1—table 4*), indicating that there was some confounding between the environmental PCAs and land-use category.

## Biodiversity differences among invertebrate taxa

Biodiversity metrics for the main insect orders (Coleoptera, Diptera, Hymenoptera, Lepidoptera, Hemiptera, and all other insects), other arthropod taxa (Collembola, mites, non-mite Arachnida, Malacostraca, myriapods), and non-arthropod phyla (Annelida, Mollusca, Nematoda, Platyhelminthes, Rotifera, and Tardigrada) that were detected showed a general trend of declining biodiversity from

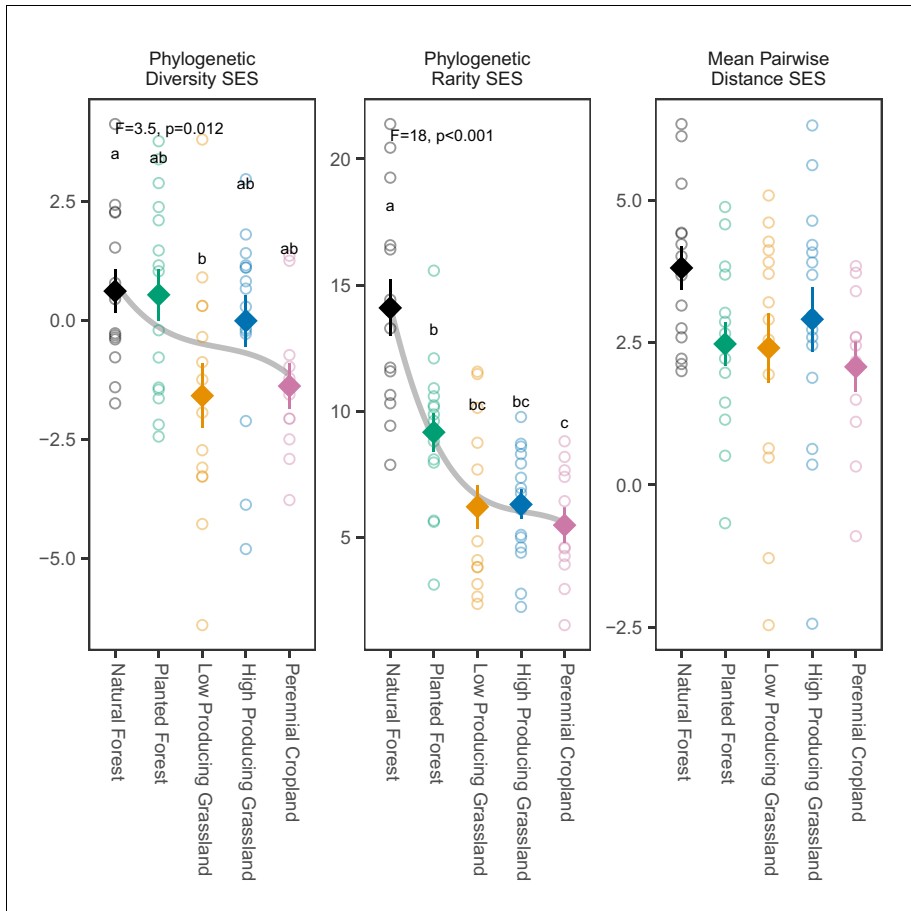

**Figure 4.** Phylogenetic biodiversity SES estimates for overall soil invertebrate communities detected in different land-use categories. Phylogenetic biodiversity SES estimates for soil invertebrate communities detected by DNA metabarcoding tend to decline from natural forest to agricultural sites, with the clearest decline shown by phylogenetic rarity SES. Diamonds and whiskers represent mean values ± standard errors, with individual data points represented by circles. ANOVA test statistics and trend splines are shown for cases with statistically significant biodiversity differences among land-use categories, with letters indicating differences between land-use categories detected by post-hoc Tukey HSD tests.

The online version of this article includes the following source data and figure supplement(s) for figure 4:

**Source data 1.** Results of ANOVA tests for differing soil invertebrate biodiversity between different land-use categories, according to three phylogenetic biodiversity SES metrics.
**Figure supplement 1.** Phylogenetic biodiversity SES estimates for soil arthropod groups detected in different land-use categories.
**Figure supplement 2.** Phylogenetic biodiversity standard effect size (SES) estimates for non-arthropod soil invertebrate phyla detected in different land-use categories.

---

forested to agricultural sites. Rarity, phylogenetic diversity, and phylogenetic rarity patterns were most consistent among different taxonomic groups (*Appendix 1—figures 7–12*), while land-use trends were most clear and consistent across taxonomic groups according to rarity and phylogenetic rarity (*Figure 3—figure supplements 1* and *2*). ANOVA tests detected significant differences among land-use categories for ten of the 17 taxonomic groups based on rarity (all insect groups, non-mites, Annelida, Nematoda, and Platyhelminthes; $F_4$ = 2.60 to 13.26, p = 0.048 to <0.001); nine groups based both on phylogenetic rarity (all insect groups except Hemiptera, mites and non-mites, Annelida, and Platyhelminthes; $F_4$ = 2.74 to 11.07, p = 0.036 to <0.001) and phylogenetic diversity (all insect groups, Annelida, Mollusca, and Nematoda; $F_4$ = 3.14 to 6.41, p = 0.047 to <0.001); eight groups based on richness (all insect groups, Nematoda, and Platyhelminthes; $F_4$ = 2.55 to 6.32, p = 0.048 to <0.001); five groups based on effective species numbers (Diptera, Hymenoptera,

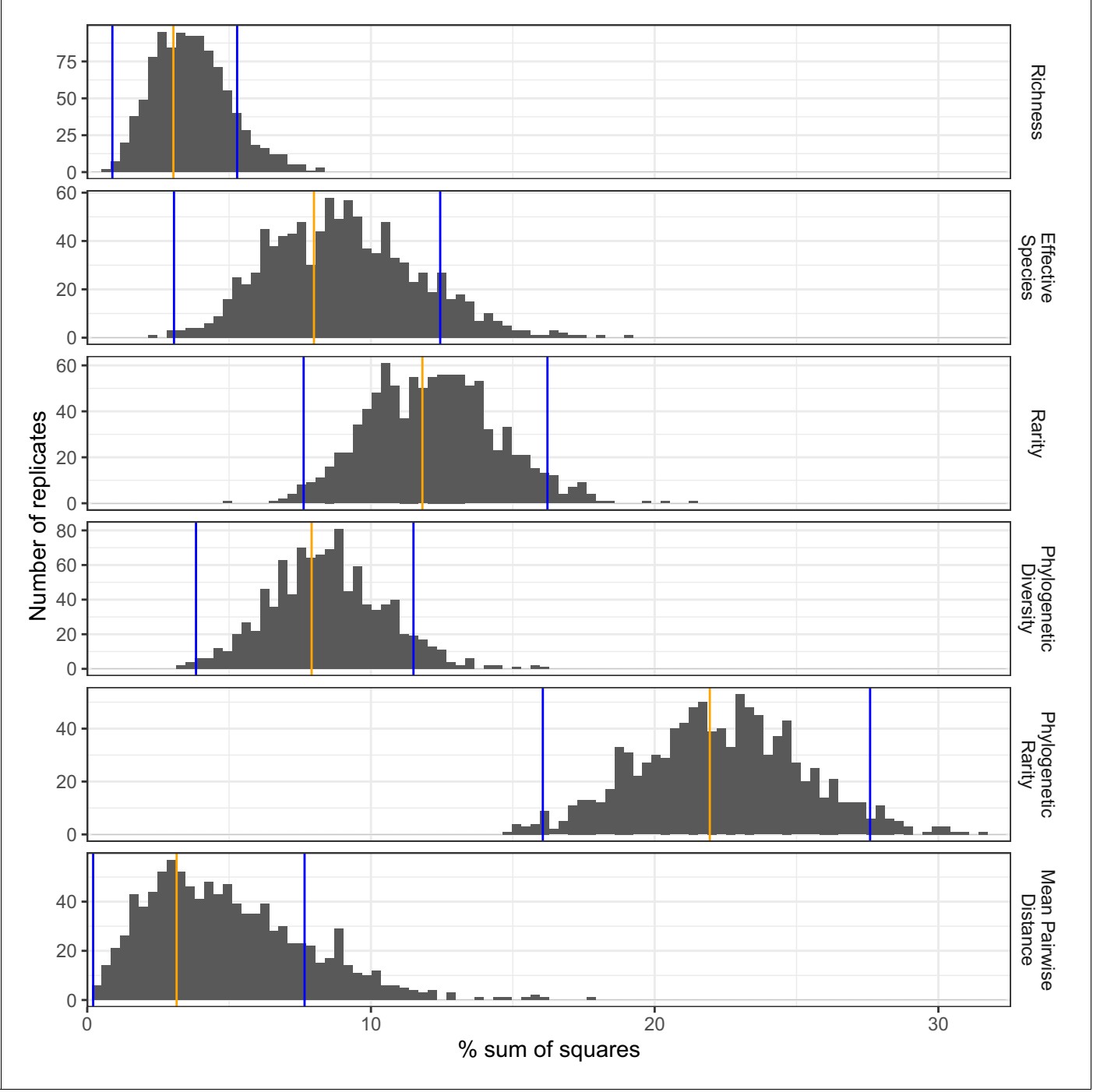

**Figure 5.** Proportions of sample variance explained by land use according to different biodiversity metrics. The proportions of sample variation (sum of squares) explained by land use were estimated for different biodiversity metrics by non-parametric bootstrapping, based on the combinations of biodiversity metric and soil invertebrate taxonomic group for which significant land-use differences were detected by ANOVA tests. Observed mean values and 95% confidence interval limits are indicated by orange and blue vertical bars, respectively.

Lepidoptera, mites, and Annelida; $F_4$ = 2.73 to 4.36, p = 0.037 to 0.004); and three groups based on mean pairwise distance differences (Hymenoptera, mites, and Rotifera; $F_4$ = 3.53 to 6.24, p = 0.012 to <0.001; *Figure 3—source data 1B*). Tests of derived land-use rank trends for each metric and taxonomic group provided concordant results, with the same groups (with few exceptions) showing significant trends for each metric (*Appendix 1—table 5*).

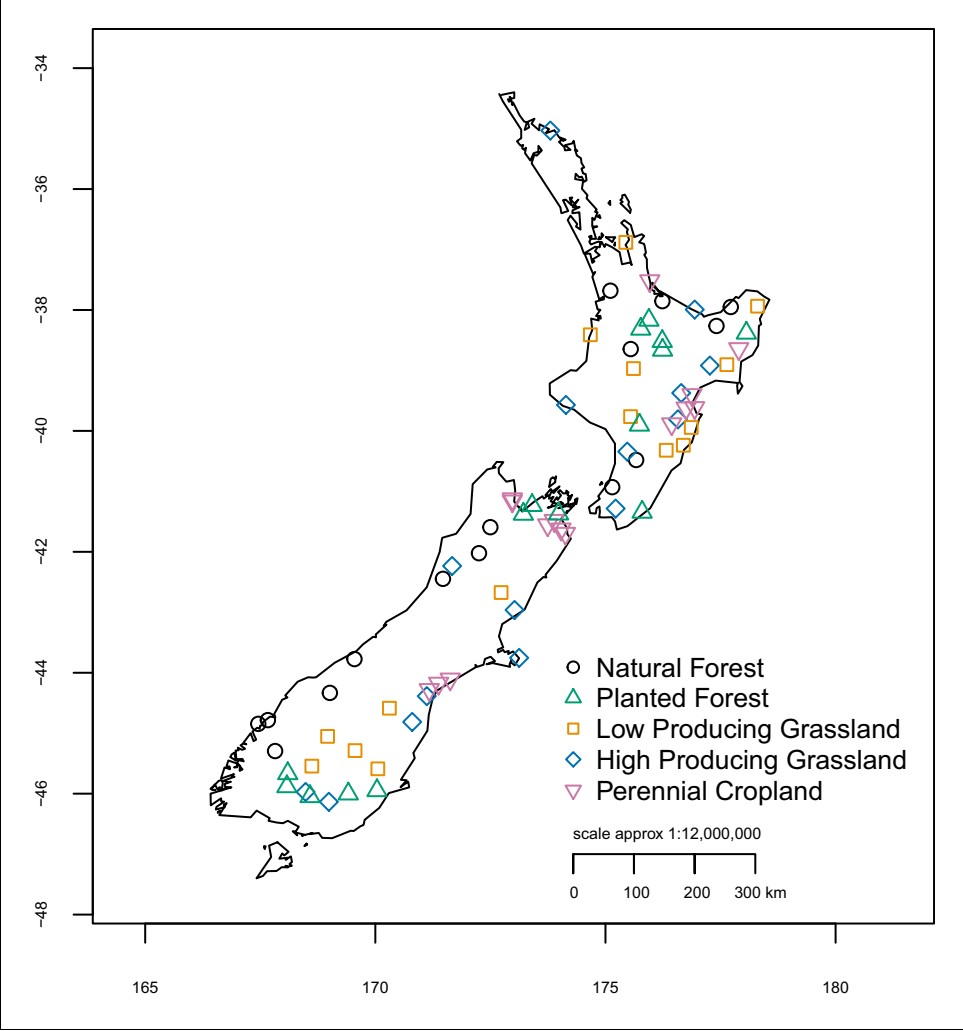

**Figure 6.** Location and land-use category of 75 sample sites. Site locations were randomly selected from a nationwide 8 km grid used for regular monitoring of native species and pests, excluding any that were >1000 m altitude and ensuring they were distributed throughout New Zealand. X- and y-axes represent longitude and latitude, respectively.

The online version of this article includes the following source data for figure 6:

**Source data 1.** Defining attributes of land-use categories.

Post-hoc Tukey HSD tests showed that biodiversity was most commonly significantly higher in natural forest compared with perennial cropland (*Figure 3—figure supplements 1* and *2*). This was observed for nine taxonomic groups based on rarity ($t_{14-23}$ = 1.92 to 7.31, *P.adj* = 0.040 to <0.001), eight groups based on phylogenetic rarity ($t_{20-28}$ = 0.054 to 1.19, *P.adj* = 0.024 to <0.001), five groups based on phylogenetic diversity ($t_{20-23}$ = 1.16 to 2.63, *P.adj* = 0.032 to <0.001), four groups based on richness ($t_{14-23}$ = 3.69 to 9.47, *P.adj* = 0.026 to <0.001), three groups based on mean pairwise distance ($t_{22-23}$ = 0.03 to 0.35, *P.adj* = 0.014 to 0.003), and just one group based on effective species numbers ($t_{25}$ = 3.00, *P.adj* = 0.012). Biodiversity was also significantly higher in natural forest compared with high-producing grassland (for two to six groups according to each of five metrics; $t_{21-27}$ = 0.02 to 6.86, *P.adj* = 0.029 to <0.001), low-producing grassland (one to five groups, four metrics; $t_{20-26}$ = 0.04 to 4.71, *P.adj* = 0.040 to <0.001), and planted forest (one to three groups, three metrics; $t_{24-27}$ = 0.64 to 4.61, *P.adj* = 0.041 to 0.007); in planted forest, low-producing grassland, or high-producing grassland compared with perennial cropland (one to two groups, two to five metrics; $t_{12-24}$ = 0.38 to 16.92, *P.adj* = 0.045 to 0.001); and in planted forest or low-producing grassland compared with high-producing grassland (one or two groups, two metrics; $t_{23-30}$ = 2.14 to 3.33,

*P.adj* = 0.036 to 0.023). All of the pairwise differences together implied a land-use category rank order of natural forest > planted forest > low producing grassland > high producing grassland > perennial cropland.

Non-parametric bootstrapping of ANOVA sum of squares values for the (non-SES) biodiversity metrics and taxonomic groups for which significant land-use differences were detected showed that phylogenetic rarity followed by (non-phylogenetic) rarity explained the largest proportions of land-use category variance across the 17 taxonomic groups, while mean pairwise distance and richness explained the least variance (*Figure 5*). A Kruskal-Wallis test detected significant differences among the biodiversity metrics (Chi square = 4782.6, *df* = 5, p<0.001), with post-hoc tests indicating that the distributions of all metrics differed significantly from each other (p<0.05).

## Phylogenetic biodiversity metric SES differences among taxa

Patterns of phylogenetic rarity SES values among land-use categories were more consistent across taxonomic groups, and with their corresponding non-SES metric patterns, than patterns of phylogenetic diversity SES and mean pairwise distance SES values (*Figure 4—figure supplements 1* and *2*). ANOVA tests detected significant differences among land-use categories for 11 of the 17 taxonomic groups based on phylogenetic rarity SES (Collembola, Coleoptera, Diptera, Lepidoptera, other insects, mites and non-mites, Annelida, Mollusca, Nematoda, and Rotifera; $F_4$ = 3.10 to 8.91, p = 0.022 to <0.001), six groups based on phylogenetic diversity SES (Hymenoptera, Lepidoptera, mites, Malacostraca, Nematoda, and Rotifera; $F_4$ = 2.76 to 7.39, p = 0.035 to <0.001); and four groups based on mean pairwise distance SES (Lepidoptera, mites, Malacostraca, and Rotifera; $F_4$ = 4.40 to 11.28, p = 0.016 to <0.001; *Figure 4—source data 1B*). All of the 11 taxonomic groups with significant phylogenetic rarity SES differences showed a consistent pattern of declining rarity from natural forest to planted forest to agricultural land-use categories. Post-hoc Tukey HSD tests detected significantly higher phylogenetic rarity SES values in natural forest (for 11 groups) and in planted forest (for four groups) compared with at least two of the agricultural land-use categories in each case ($t_{22-28}$ = −1.73 to −3.77, *P.adj* = 0.047 to <0.001). In contrast, only two groups (mites and Rotifera) showed this pattern based on either phylogenetic diversity SES ($t_{22-28}$ = −1.52 to −3.15, *P. adj* = 0.031 to <0.001) or mean pairwise distance SES values ($t_{22-28}$ = −1.83 to −2.89, *P.adj* = 0.047 to <0.001). Otherwise, Lepidoptera phylogenetic diversity SES values were significantly lower in both planted forest and high-producing grassland compared with both natural forest and perennial cropland ($t_{21-27}$ = −1.07 to −1.44, *P.adj* = 0.035 to 0.004), whereas Hymenoptera, Malacostraca and Nematoda phylogenetic diversity SES values were higher in one or more of the anthropogenic land use categories compared with natural forest ($t_{3-28}$ = 1.46 to 2.87, *P.adj* = 0.020 to 0.005). Patterns of mean pairwise distance SES values across land use categories and taxonomic groups closely matched those observed for phylogenetic diversity SES values (except significant differences among land-use categories were not detected for Hymenoptera or Nematoda).

## Discussion

This research provides clear evidence of adverse impacts of agricultural land use upon soil invertebrate communities. Effects of land use on biological communities are usually measured as shifts in species richness. However, rarity metrics were much more sensitive to land use and more consistent among taxa than richness or effective species numbers in our study, suggesting that the latter metrics may underestimate land-use impacts on biodiversity. Rarity is a function of the number of species with limited distributions or narrow habitat specificity. These rare species can have important roles in ecosystem processes (*Dee et al., 2019*; *Leitão et al., 2016*; *Lyons et al., 2005*), and are inherently more vulnerable to extinction. Overlooking species rarity, as richness does, therefore obscures the effects of different land uses on communities, with potential detrimental consequences for the function and resilience of ecosystems. Our results suggest that efficient DNA-based measurement of plot-level rarity improves our understanding of rare species occurrence and provides an effective basis for incorporating soil invertebrates into conservation planning.

Rare species include not only habitat specialists, but also transient and conditionally rare taxa. It is possible that OTUs that were rare in this study may be more common in locations not sampled. Nonetheless, our observation that patterns of rarity among land-use categories were the most consistent among different taxa suggests that rarity is an ecologically meaningful measure of ecosystem

biodiversity. This is supported by prior studies suggesting that rare species are particularly sensitive to ecosystem change. For example, rare plant and fungal species appear to be particularly sensitive to changes in environment (*Avis et al., 2008*; *Dickie et al., 2009*; *Dickie and Reich, 2005*; *McIntyre and Lavorel, 1994*), and pollinating insect losses are concentrated among rare species (*Powney et al., 2019*). Furthermore, most of the terrestrial invertebrate species currently considered to be at risk or threat of extinction in New Zealand are naturally uncommon (*Stringer and Hitch-mough, 2012*).

Phylogenetic diversity – and especially phylogenetic rarity – explained larger proportions of land-use variance across taxa than their non-phylogenetic counterparts, and phylogenetic rarity was over-all the most sensitive metric to land-use differences. Phylogenetic metrics incorporate evolutionary and functional aspects of biodiversity (*Faith, 1992*; *Faith, 2015*; *Mazel et al., 2018*). New Zealand has a long history of geographic isolation and glaciation, reflected by the presence of many deeply divergent invertebrate lineages (*Buckley et al., 2015*; *Trewick et al., 2011*). The high levels of inver-tebrate phylogenetic rarity in natural forest sites likely reflects assemblages of long-present soil invertebrates that are highly adapted to these habitats, but ill-suited to the modified land-use types included in the study. These trends might differ in regions with greater connectivity, longer-term agriculture, and different geological history. Phylogenetic diversity SES and mean pairwise distance SES values showed different evidence of land-use effects compared with their non-SES counterparts, suggesting, for example, that Lepidoptera, mite and Rotifera communities are less dispersed, sug-gesting loss of lineages, in agricultural sites compared with forest habitats. In contrast, Malacostraca communities appear to be under-dispersed in natural forest sites, and to gain lineages due to anthropogenic land use. Phylogenetic rarity SES values further support the finding of consistently reduced rarity in agricultural sites, independent of species richness effects. Together, these observa-tions indicate that phylogenetic information provides additional insights into soil invertebrate biodi-versity patterns, as has been observed for other groups (*González-Orozco et al., 2015*; *Mishler et al., 2014*).

## Land-use impacts

The low beta diversity, heterogeneity, and rarity values detected in agricultural sites, and the overlap of samples from these sites in MDS ordinations, together strongly imply that these habitats tend to have relatively similar assemblages of species across locations. Agricultural practices have effects at a wide range of scales, from local-scale use of chemical fertilisers and pesticides to landscape-scale habitat simplification (*Tscharntke et al., 2005*). Together these factors lead to homogenisation of communities and functions among sites, in which specialists in diverse natural communities are replaced by a smaller number of generalists that thrive in anthropogenic habitats (*Börschig et al., 2013*; *Clavel et al., 2011*; *Gámez-Virués et al., 2015*; *McKinney and Lockwood, 1999*; *Smart et al., 2006*).

In contrast to the agricultural sites, the high diversity and rarity observed in natural forest sites indicates that these habitats tend to have richer and more unique assemblages of species. Forested sites tend to have greater physical habitat complexity and heterogeneity, providing more varied resources and niches for diverse communities including various specialists (*Jonsson et al., 2009*; *Stein et al., 2014*). Furthermore, natural forest habitats tend to be more disconnected, and located in more rugged and less accessible areas than agricultural sites, with more physical barriers to limit the dispersal of invertebrate fauna. Consequently, the distinct assemblages detected in natural for-est sites are likely to reflect natural historical biogeographic distribution and evolutionary processes (*Buckley et al., 2015*; *Trewick et al., 2011*).

Despite their varying sensitivity, most metrics of rarity and diversity (not mean pairwise distance, phylogenetic diversity SES, or mean pairwise distance SES) showed a consistent trend of lower biodi-versity in agricultural land-use categories than in forested land-use categories. Further, while not all taxa showed significant evidence of declining biodiversity in relation to agricultural land use, no taxa responded positively. Many taxa not showing significant biodiversity declines had few species (e.g. myriapods, Malacostraca and tardigrades), suggesting there was insufficient data to infer land-use differences. Among the most species-rich groups that did not show significant declines (collembola, mites and rotifers), many of the diversity metrics were nonetheless lowest in grassland or perennial cropland sites, suggesting that while these groups may be more resilient to impacts of agricultural land use than others, the general trend was similar. These biodiversity declines are in contrast to

previous research that suggested soil fauna are resilient to grassland intensification (*Gossner et al., 2016*), likely because our study encompasses a broader range of land-use types. While it is likely that spatial and environmental factors associated with particular land uses contribute to these patterns, the fact that land use explained additional variation of richness and rarity metrics after these factors were statistically accounted for strongly indicates an independent role of land management practices.

While rarity and phylogenetic rarity metrics showed the most consistent responses across land-use categories, the rank order of land-use categories implied by these (and other) metrics were not easily predicted prior to measurement. Planted forests, which were predominantly *Pinus radiata* plantations, are sometimes perceived as being biologically depauperate, while low-producing grasslands are frequently perceived as semi-natural in New Zealand (*Hobbs et al., 2006*). Despite this, we found rarity and diversity in planted forest sites to be similar to those in low-producing grassland sites and higher than those in high-producing grassland or perennial cropland sites, consistent with suggestions that plantations can play an important role in insect biodiversity conservation (*Pawson et al., 2009*; *Pawson et al., 2010*). Similarly, high-productivity grasslands are often perceived as a more severe land use than perennial cropland due to high homogeneity of vegetation cover, low habitat complexity, and high fertiliser use. Nonetheless, our data suggest perennial cropland supports the lowest levels of invertebrate diversity and rarity of any of the measured land-use categories. This may reflect high chemical input in and intensive management of fruit production systems (*Manktelow et al., 2005*).

Overall, our results suggest pervasive impacts of agricultural land use upon soil invertebrate communities, with likely adverse consequences for ecosystem services. This adds to widespread evidence of declines in invertebrate biomass and diversity in response to anthropogenic land-use change and habitat loss (*Attwood et al., 2008*; *Hallmann et al., 2017*; *Hendrickx et al., 2007*; *Powney et al., 2019*), and suggests that efforts to conserve and restore soil invertebrate communities may be needed.

## Conservation implications

Invertebrates tend to be neglected by conservation initiatives, due to the challenges of determining their identities, functions, and distributions (*Leandro et al., 2017*). Indirect preservation of communities via flagship or umbrella species protection schemes tends to be ineffective (*Andelman and Fagan, 2000*; *Oberprieler et al., 2019*; *Schuldt and Assmann, 2010*), and similarly, biomonitoring based on individual species is problematic. By allowing the efficient assessment of invertebrate community composition and distribution across large spatial scales, DNA metabarcoding methods may enable more informative biomonitoring and improved targeting of conservation initiatives based on multiple invertebrate taxa, if not entire invertebrate communities. While rarity and phylogenetic rarity were the most informative metrics of community change in this case, it is likely that consideration of these alongside richness and phylogenetic measures of diversity would provide the most comprehensive information for purposes such as biomonitoring and conservation planning (*Fleishman et al., 2006*). Our results suggest that conserving a network of sites with high invertebrate diversity and rarity would preserve a diverse assemblage of species, communities, and functional traits, thus providing resilience of communities and ecosystem processes to environmental changes (*Balvanera et al., 2006*; *Yachi and Loreau, 1999*). While diversity and rarity was typically highest in our natural forest sites (of which many are protected), certain grassland and cropland sites with unusually high rarity values (outliers on *Figure 3*) might be logical targets for further investigation and potential incorporation into conservation initiatives.

In conclusion, our analysis of soil invertebrate biodiversity across land-use categories at a national scale shows that most soil invertebrate taxa have consistent rarity responses to land use, and that agricultural land use tends to cause the homogenisation and loss of soil invertebrate biodiversity. This research adds to evidence of widespread impacts of anthropogenic land use on invertebrate biodiversity, but also implies that these impacts may have been underestimated due to a widespread emphasis on species richness. DNA metabarcoding methods offer an efficient basis for measuring the diversity and rarity of invertebrate communities at large scales. Incorporating this information into conservation schemes would enable the protection of a broader range of biodiversity and enhance the preservation of terrestrial ecosystems.

# Materials and methods

## Key resources table

| Reagent type (species) or resource | Designation | Source or reference | Identifiers | Additional information |
|---|---|---|---|---|
| Sequence-based reagent | mlCOIintF | DOI: 10.1186/1742-9994-10-34 | | GGWACWGGWTGAACWGTWTAYCCYCC |
| Sequence-based reagent | HCO2198 | PMID:7881515 | | TAAACTTCAGGGTGACCAAAAAATCA |
| Commercial assay or kit | NucleoSpin Tissue kit | Macherey-Nagel | 740741.4 | |
| Software, algorithm | cutadapt | https://github.com/marcelm/cutadapt | v 1.11 | |
| Software, algorithm | USEARCH | https://www.drive5.com/usearch/ | v 9.0.2132_i86linux32 | |
| Software, algorithm | VSEARCH | https://github.com/torognes/vsearch | v 2.4.0 | |
| Software, algorithm | R | https://www.r-project.org/ | v 3.52 | |
| Software, algorithm | phylo.endemism | https://davidnipperess.blogspot.com/2012/07/phyloendemism-r-function-for.html | | |

## Sample collection

Soil invertebrate communities were sampled from a total of 75 sites distributed across five different major land-use categories throughout New Zealand (*Figure 6*), during dry weather between November 2014 and March 2015. The five land-use categories (natural forest, planted forest, low-producing grassland, high-producing grassland, and perennial cropland) represent differing states of anthropogenic modification (*Figure 6—source data 1*). The site locations were selected from a nationwide 8 km grid used for regular monitoring of native species and pests. For each land-use category, 15 replicate sites were randomly selected from the nationwide monitoring grid, excluding any that were >1000 m altitude and ensuring they were distributed across the length of New Zealand (*Makiola et al., 2019*). At each site, a 20 m × 20 m plot was established according to a standardised protocol (*Hurst and Allen, 2007*). Twenty-four soil cores were collected within each plot on a regular grid (min 3.54 m distance between cores) to a depth of 15 cm using a sterile corer (5.08 cm diameter), following *Wood et al. (2017)*. Surface litter was removed prior to coring. The 24 soil cores were pooled together, homogenised, and stored at 4°C until laboratory processing. Invertebrates were extracted from a one-litre subsample of homogenised soil material from each site using Berlese-Tullgren funnels and stored in ethanol until DNA extraction.

The altitude and latitude of plots were determined from topographic maps. Soil chemistry variables (pH, C, N, C:N ratio, Olsen P, Total P, Ca, Mg, K, Na, cation exchange capacity, base saturation) were determined for each plot according to *Orwin et al. (2016)* and *Wood et al. (2017)*.

## Molecular laboratory procedures

Bulk invertebrate concentrates were centrifuged for three minutes at 2,500 rpm (1258 rcf), after which ethanol was removed until <5 ml remained. The concentrates were then transferred into 5 ml tubes and homogenised with eight steel balls in a bead mill operated at 15 Hz for six intervals of 20 s each. A 1.5 ml aliquot of homogenised invertebrate concentrate from each sample was removed into a 1.5 ml microtube and centrifuged for one minute at 13,000 rpm (11,337 rcf), after which any ethanol was removed. The pelleted material was resuspended in purified water, re-centrifuged as before, then resuspended in 200 μl digestion buffer (10 mM Tris buffer, 10 mM NaCl, 5 mM CaCl2, 2.5 mM EDTA, 2% SDS, 0.04 M dithiothreitol, and 0.1 M proteinase K) with vortexing, and incubated overnight at 56 °C with shaking at 450 rpm (*Campos and Gilbert, 2012*). DNA was extracted from the digested samples using a Macherey-Nagel NucleoSpin Tissue kit (MACHEREY-NAGEL GmbH and Co. KG, Düren, Germany), omitting sample lysis steps but otherwise according to the

manufacturer's directions, with a JANUS workstation laboratory robot (PerkinElmer, Waltham, MA, USA). The DNA concentration was quantified in each extract using an Invitrogen Quant-iT PicoGreen dsDNA quantitation assay kit (Thermo Fisher Scientific, Waltham, MA USA), and standardised across samples to 3 ng/μl.

COI barcodes were amplified by PCR from each sample using metazoan-targeted primers mlCOIintF (5′-GGWACWGGWTGAACWGTWTAYCCYCC-3′) (*Leray et al., 2013*) and HCO2198 (5′-TAAACTTCAGGGTGACCAAAAAATCA-3′) (*Folmer et al., 1994*), which were respectively modified at their 5′ ends with the linker sequences 5′-TCGTCGGCAGCGTC-3′ and 5′-GTCTCGTGGGCTCGG-3′. PCRs were carried out in 20 μl volumes, containing 200 nM of the forward and reverse COI primers, 0.2 mM of each dNTP, 1.5 mM MgCl$_2$, 2 μg rabbit serum albumin, 0.5 U KAPA Plant 3G enzyme (Kapa Biosystems, Wilmington, MA, USA), and 2 μl (6 ng) DNA template. The PCR amplification protocol was 95 ℃ for 3 min; 35 cycles of 95 ℃ for 20 s, 52 ℃ for 15 s, and 72 ℃ for 30 s; and 1 min at 72 ℃. Illumina sequencing adapters and sample-specific barcodes were added to the COI amplicons in a second round of PCR, carried out in 25 μl volumes containing the same reagents and concentrations as the first PCR, except for Illumina-tagged sequencing adaptors instead of COI primers, and 2 μl of the first PCR amplicon as template. The second-round PCR amplification protocol was 95 ℃ for 3 min; five cycles of 95 ℃ for 20 s, 54 ℃ for 15 s, and 72 ℃ for 30 s; and 1 min at 72 ℃. The resulting libraries were purified and size-selected using a Pippin Prep system (Sage Science, Beverly, MA, USA), to remove primer dimers and high molecular weight DNA, quantified, pooled, and sequenced on an Illumina MiSeq system with a 2 × 250 sequencing kit at the Australian Genome Research Facility Ltd.

## Bioinformatic processing

Demultiplexed forward and reverse DNA reads were merged and relabelled by sample using USEARCH (*Edgar, 2013*). Linker sequences and primers were trimmed from the merged sequences using cutadapt (*Martin, 2011*). The trimmed sequences were quality filtered to remove any with >1 maximum expected errors and dereplicated using VSEARCH (*Rognes et al., 2016*). Non-singleton sequences (i.e. those represented by at least two identical sequences) were clustered into OTUs at a sequence identity threshold of 97% and simultaneously filtered for chimeras using the UPARSE algorithm in USEARCH (*Edgar, 2013*). OTU abundance was inferred by mapping the trimmed sequences back to the OTU centroid sequences at a sequence identity threshold of 97%. The OTUs were assigned a taxonomic identity using the RDP Naïve Bayesian classifier (*Wang et al., 2007*) in combination with an RDP-formatted animal mitochondrial COI sequence database (*Porter and Hajibabaei, 2018*), which includes bacterial, fungal, and protist COI sequences to enable the detection of non-metazoan OTUs. We excluded any OTUs that were not identified as belonging to an expected terrestrial invertebrate phylum.

## Biodiversity analyses and statistics

Data analyses were carried out using R version 3.5.1 (*R Development Core Team, 2016*) and RStudio (*RStudio team, 2015*). Extraction blanks, negative and positive controls were examined for contamination. Tag jumping (*Schnell et al., 2015*) was accounted for by using a regression of contaminant abundances versus the maximum of total abundances in all other samples, after which the coefficient estimate for the 90th quantile regression was used to subtract that many sequences from the abundances of all OTUs (*Makiola et al., 2019*).

Comparisons of multivariate community composition and homogeneity between land-use categories were carried out for the overall terrestrial invertebrate dataset and the main terrestrial invertebrate phyla detected using the R package vegan v2.4–3 (*Oksanen et al., 2017*). Non-metric MDS ordinations and PERMANOVA tests for community composition differences among land uses were based on the Jaccard distance metric and presence/absence data. Any samples with unusually low sequence abundance (defined as less than 5% of the mean sequence abundance per sample for a given phylum) were excluded from MDS ordinations. For the Mollusca-based MDS ordination, one further sample that resulted in an uninterpretable plot was excluded. To test for homogenisation effects of land use, multivariate homogeneity of sample dispersions was determined for each land-use category and compared between categories using the function betadisper in the R package vegan. Similarly, mean pairwise beta diversity and phylogenetic beta diversity (UniFrac distances;

*Lozupone and Knight, 2005*) were calculated for each land-use category, and compared between land-use categories using ANOVA and post-hoc Tukey HSD tests. Heatmaps of relative OTU abundance and distribution among sites were generated using phyloSeq (*McMurdie and Holmes, 2013*), for the 1000 terrestrial invertebrate OTUs with the highest proportional abundances across sites.

Biodiversity estimates were calculated for each sample based on the overall terrestrial invertebrate communities, and for each of the main invertebrate groups detected, in such a way that all terrestrial invertebrate OTUs were represented: (1) the dominant insect orders detected (Coleoptera, Diptera, Hymenoptera, Lepidoptera, and Hemiptera, each represented by >150 OTUs); a further 18 insect orders represented by 1 to 36 OTUs were considered as a single pooled group ('other insects'); (2) non-insect arthropod groups (non-mite Arachnida, mites, Collembola, Malacostraca, myriapods); and (3) non-arthropod phyla (Annelida, Mollusca, Nematoda, Platyhelminthes, Rotifera, and Tardigrada). Because many OTUs were only found in a single site, biodiversity estimates were also calculated with these OTUs excluded, to check whether this affected the results. Species richness and effective species numbers (exponential of Shannon entropy; Jost 2006), were calculated for each invertebrate group using the R packages vegan v2.4–3 (*Oksanen et al., 2017*) and vegetarian v1.2 (*Charney, 2012*) respectively. To calculate rarity, a weighting factor ($w$) was determined for each OTU as the reciprocal of its occurrence across all samples (regardless of land use), so that $w = 1$ for OTUs that occur in only in a single sample, and $w$ approaches zero for OTUs that occur in many samples. For each sample, values of $w$ were then summed for all OTUs occurring in that sample. In other words, rarity represents the number of OTUs per sample adjusted for their occurrence across all samples (*Kier et al., 2009*; *Kier and Barthlott, 2001*).

To calculate phylogenetic diversity, phylogenetic rarity, and mean pairwise distance, OTU sequences were aligned using MAFFT v7 (*Katoh and Standley, 2013*), and phylogenetic trees constructed. Initially, phylogenetic trees were constructed separately for each phylum using both FastTree 2 (*Price et al., 2010*) and RAxML v8 (*Stamatakis, 2014*), and for the overall invertebrates using FastTree 2 (construction of the overall invertebrates tree using RAxML failed). As phylum-level trees based on each method and the overall tree pruned to each phylum yielded similar results, the overall tree was used for estimation of phylogenetic biodiversity metrics per sample and taxonomic group. Phylogenetic diversity, in the form of total branch length per sample, and mean pairwise distance were calculated for each taxonomic group using the R package Picante (*Kembel et al., 2010*). Phylogenetic rarity, in the form of the branch length unique to each sample (based on occurrences across all samples), was calculated for each taxonomic group and sample according to *Rosauer et al. (2009)* using the R function *phylo.endemism* (*Niperess, 2010*). In addition, standardised effect size values were calculated for each of the phylogenetic metrics, by comparing observed values per site to a null distribution generated by 999 randomisations of the data using a regional null model (*Kembel et al., 2010*; *Miller et al., 2017*).

ANOVA was used to test for significant differences among mean biodiversity values between land-uses, for overall invertebrate communities and for each of the taxonomic groups, based on each of the biodiversity metrics. We considered land use as an unordered categorical factor in these tests, because we had no a priori expectation about the relative intensity or impact of all five land uses. Any statistically significant ANOVA tests were followed with post hoc two-sided Tukey HSD tests to identify significant pairwise differences among land-use categories. Subsequently, based on our observed rank order of land uses, we derived a numeric rank of 1 to 5 in the order natural forest > planted forest > low producing grassland > high producing grassland > perennial cropland. We refer to this numeric rank as derived land-use rank (DLUR in tables), to make clear that it is derived from our observed results, rather than on any a priori hypothesis as to which land uses might be considered more intense than others. We tested whether this provided the same conclusions as treating land use as a categorical factor for each metric and taxonomic group. We also included DLUR in a further ANOVA test for biodiversity and taxonomic group differences, to test whether different taxonomic groups showed the same patterns.

We also investigated whether environmental covariates might explain biodiversity trends of overall soil invertebrate communities. To do so, we carried out ANOVA tests for effects of spatial variables (latitude and altitude) plus land-use category effects on overall biodiversity estimates for each metric. In addition, we generated a PCA based on spatial and soil chemistry variables. We then tested whether the most important PCA components, plus land-use category, had significant effects on overall biodiversity estimates for each metric.

To investigate whether the biodiversity metrics differed in their sensitivity to land use, non-parametric bootstrapping stratified by taxonomic group with 999 replicates was used to estimate the proportion of variance attributable to land-use effects with 95% confidence intervals, across the set of taxonomic groups and metrics for which significant land-use differences were detected by ANOVA. These results were plotted as a histogram and compared between metrics using a non-parametric Kruskal-Wallis test.

## Acknowledgements

We thank landowners, and the New Zealand Department of Conservation, for allowing access to sampling sites, Manaaki Whenua – Landcare Research staff, especially Chris Morse, Larry Burrows, Thomas Easdale, Karen Boot, Nicola Bolstridge, Hamish Maul, and Carina Davis, for their extensive support in the field and laboratory, and peer reviewers for their helpful insights. This research was funded by the New Zealand Ministry of Business, Innovation and Employment (MBIE Contract No. C09X1411) via Manaaki Whenua – Landcare Research.

## Additional information

### Funding

| Funder | Grant reference number | Author |
|---|---|---|
| Ministry of Business, Innovation and Employment | MBIE Contract No. C09X1411 | Kate H Orwin<br>Robert J Holdaway<br>Jamie R Wood<br>Ian A Dickie |

The funders had no role in study design, data collection and interpretation, or the decision to submit the work for publication.

### Author contributions

Andrew Dopheide, Conceptualization, Formal analysis, Software, Bioinformatics, Data curation, Visualization, Writing - original draft, Writing - reviewing and editing; Andreas Makiola, Investigation, Fieldwork, Writing - reviewing and editing; Kate H Orwin, Conceptualization, Funding acquisition, Writing - reviewing and editing; Robert J Holdaway, Conceptualization, Funding acquisition, Investigation, Fieldwork, Writing - reviewing and editing; Jamie R Wood, Funding acquisition, Investigation, Molecular analysis, Writing - reviewing and editing; Ian A Dickie, Conceptualization, Funding acquisition, Investigation, Fieldwork, Formal analysis, Software, Bioinformatics, Writing - reviewing and editing

### Author ORCIDs

Andrew Dopheide https://orcid.org/0000-0003-1554-9832
Andreas Makiola https://orcid.org/0000-0002-9611-9238
Kate H Orwin https://orcid.org/0000-0003-0906-728X
Jamie R Wood https://orcid.org/0000-0001-8008-6083
Ian A Dickie https://orcid.org/0000-0002-2740-2128

### Decision letter and Author response

Decision letter https://doi.org/10.7554/eLife.52787.sa1
Author response https://doi.org/10.7554/eLife.52787.sa2

## Additional files

### Supplementary files

• Transparent reporting form

## Data availability

Sequence data, metadata, processed data, bioinformatic processing and analysis code used to generate the results in the manuscript (with one exception, detailed below) are deposited in the Manaaki Whenua-Landcare Research DataStore, accessible at: https://doi.org/10.7931/w3j3-5v40. Our sample sites include many Māori and/or privately-owned locations. We have have removed site location details from our metadata out of respect for concerns of Māori and other landowners. The removal of site location details precludes recreation of the map of sample site locations (Figure 6) and analyses of latitude effects, but otherwise has no impact on our results.

The following dataset was generated:

| Author(s) | Year | Dataset title | Dataset URL | Database and Identifier |
|---|---|---|---|---|
| Dopheide A, Makiola A, Orwin KH, Holdaway R, Wood JR, Dickie I | 2019 | Land use impacts on soil invertebrate biodiversity | https://doi.org/10.7931/w3j3-5v40 | Manaaki Whenua - Landcare Research DataStore, 10.7931/w3j3-5v40 |

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

# Appendix 1

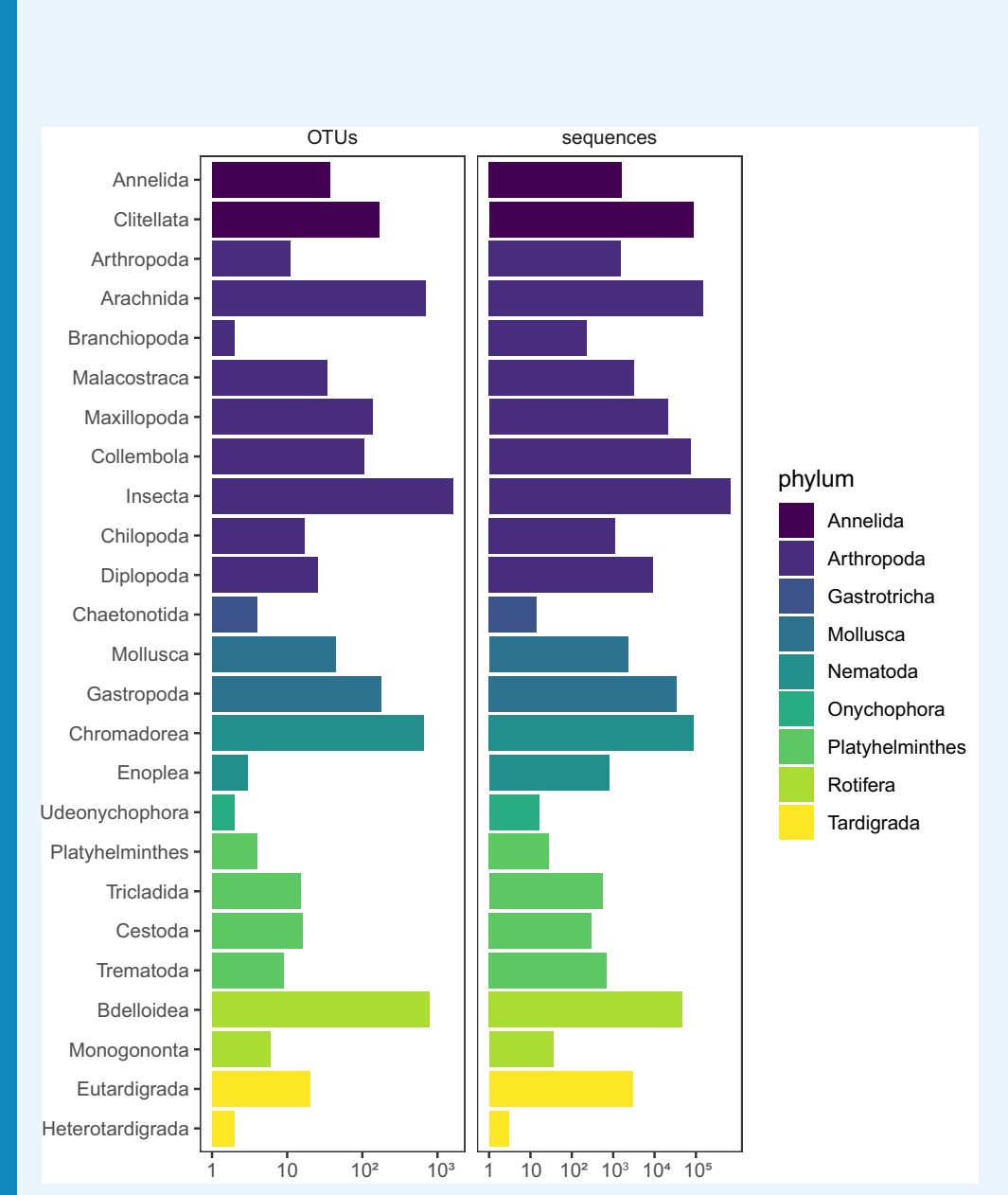

**Appendix 1—figure 1.** Taxonomic composition of invertebrate OTUs and sequences. Phylum and class-level taxonomic composition of terrestrial invertebrate OTUs detected in soil samples from 75 sites distributed across five land-use categories.

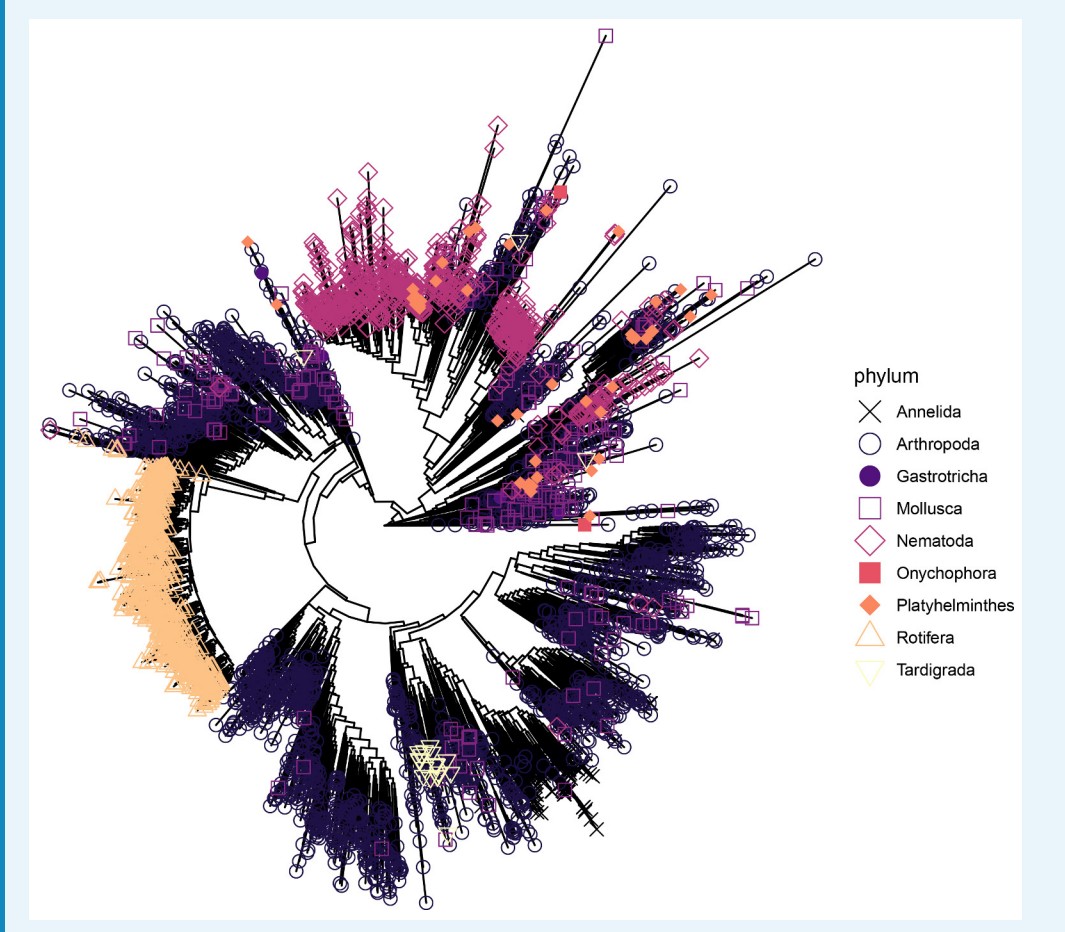

**Appendix 1—figure 2.** A phylogeny of terrestrial invertebrate COI OTU sequences detected in soil samples from 75 sites distributed across five land-use categories.

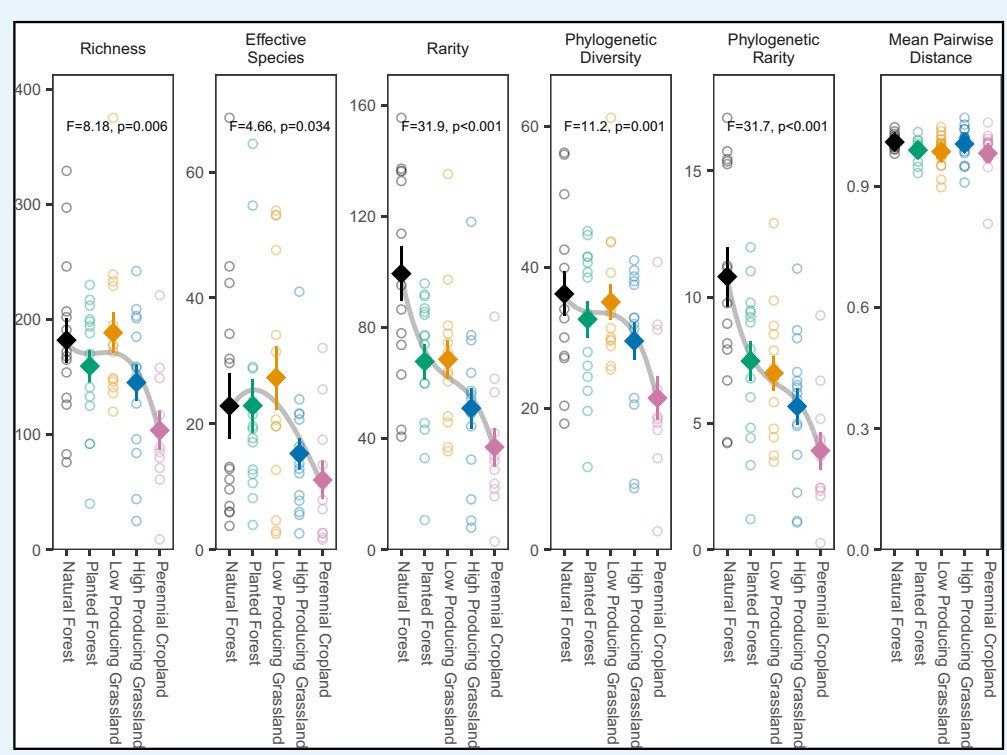

**Appendix 1—figure 3.** Biodiversity estimates for overall soil invertebrate communities detected in different land-use categories, with species detected in a single site excluded. Diamonds and whiskers represent mean values ± standard errors, with individual data points represented by circles. ANOVA test statistics and trend splines are shown for cases with statistically significant biodiversity differences among land-use categories, with letters indicating differences between land-use categories detected by post-hoc Tukey HSD tests.

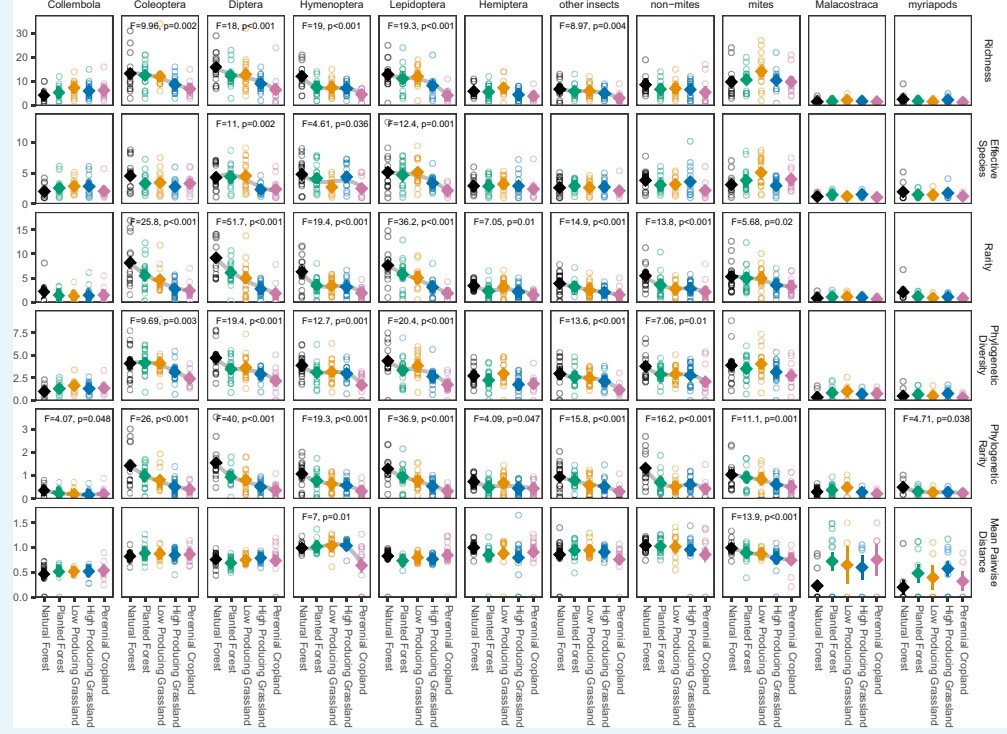

**Appendix 1—figure 4.** Biodiversity estimates for soil arthropod groups in different land-use

categories, with species detected in a single site excluded. 'Other insects' consists of all insect orders other than Coleoptera, Diptera, Hemiptera, Hymenoptera, and Lepidoptera. 'Non-mites' consist of Araneae, Opiliones, and Pseudoscorpiones. Diamonds and whiskers represent mean values ± standard errors, with individual data points represented by circles. ANOVA test statistics and trend splines are shown for cases with statistically significant biodiversity differences among land-use categories, with letters indicating differences between land-use categories detected by post-hoc Tukey HSD tests.

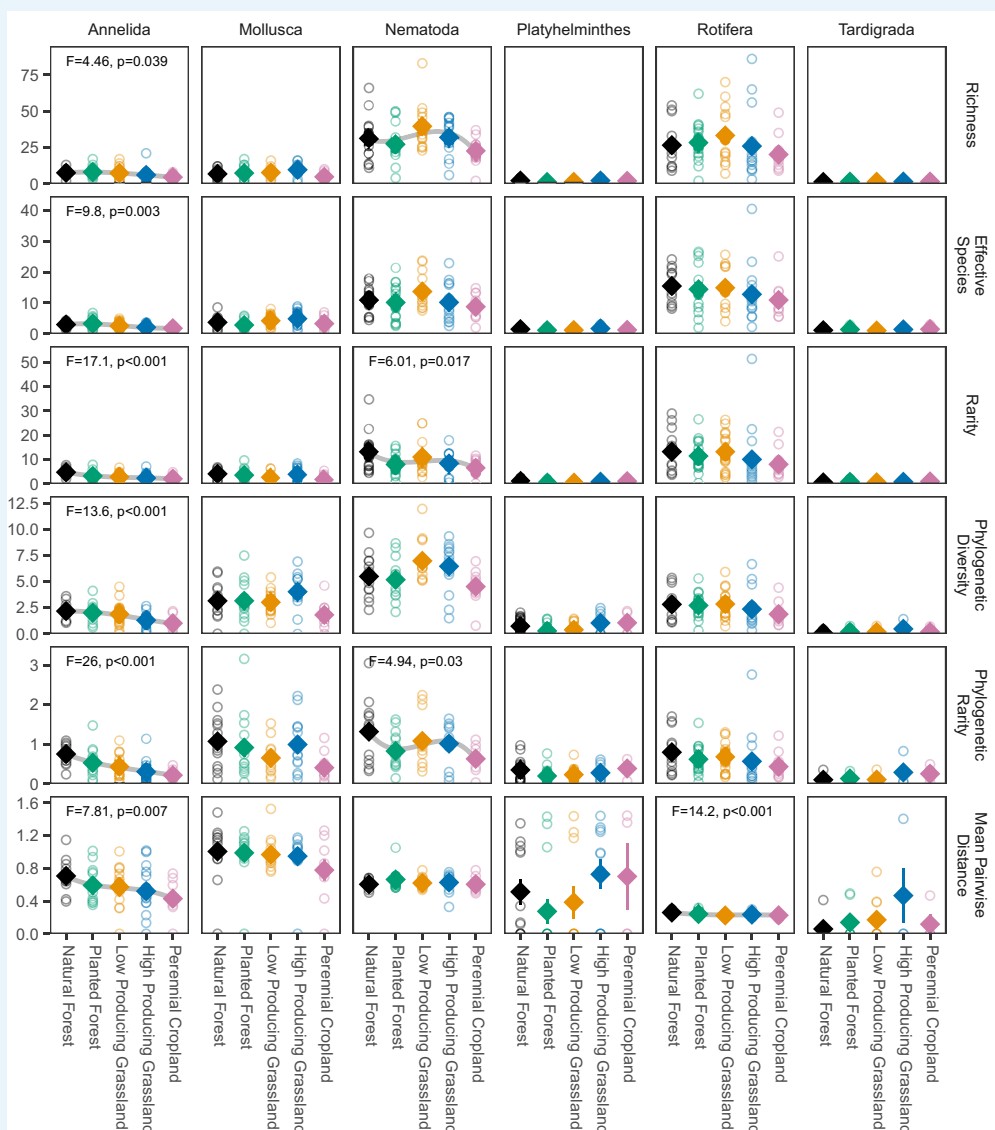

**Appendix 1—figure 5.** Biodiversity estimates for non-arthropod soil invertebrate phyla in different land-use categories, with species detected in a single site excluded. Diamonds and whiskers represent mean values ± standard errors, with individual data points represented by circles. ANOVA test statistics and trend splines are shown for cases with statistically significant biodiversity differences among land-use categories, with letters indicating differences between land-use categories detected by post-hoc Tukey HSD tests.

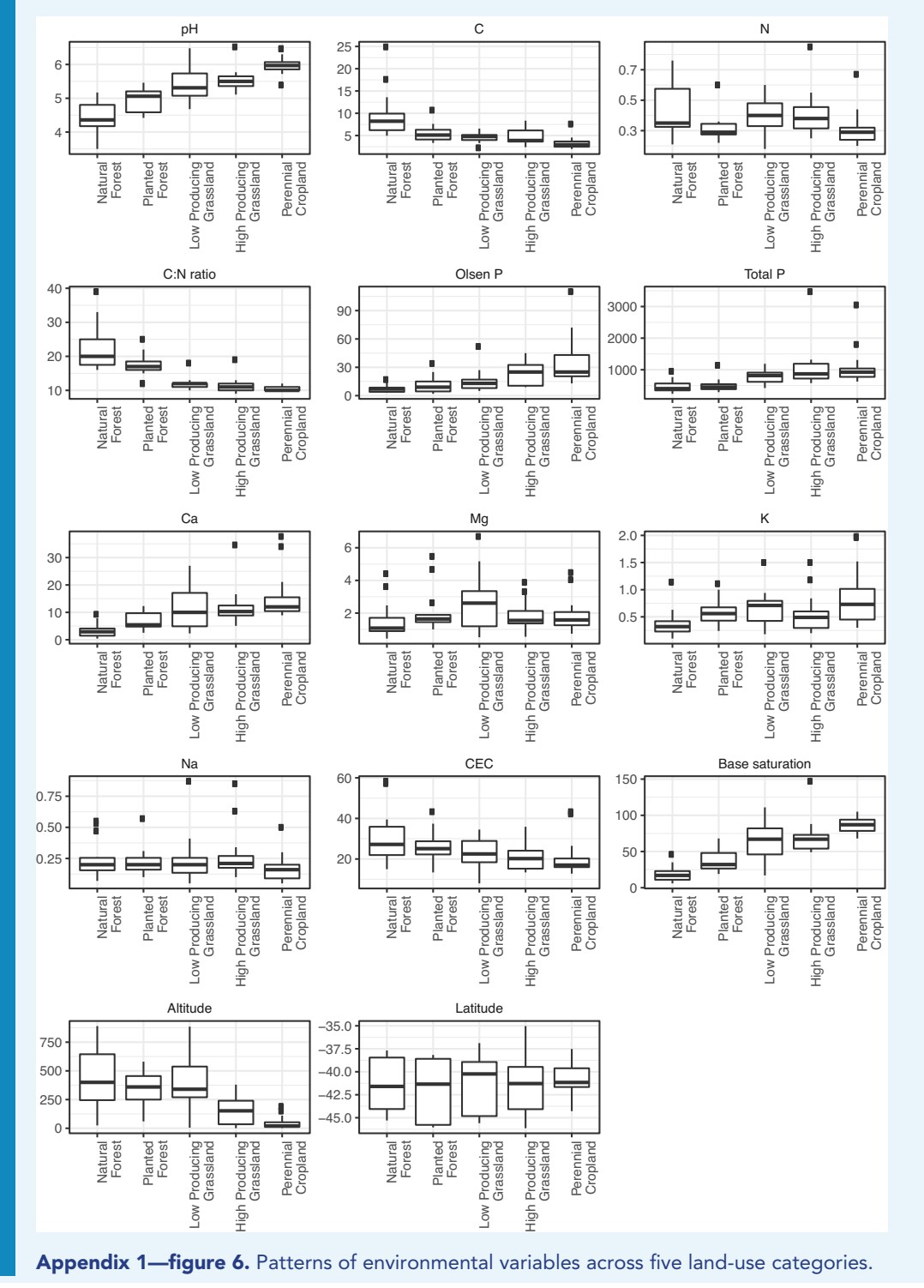

**Appendix 1—figure 6.** Patterns of environmental variables across five land-use categories.

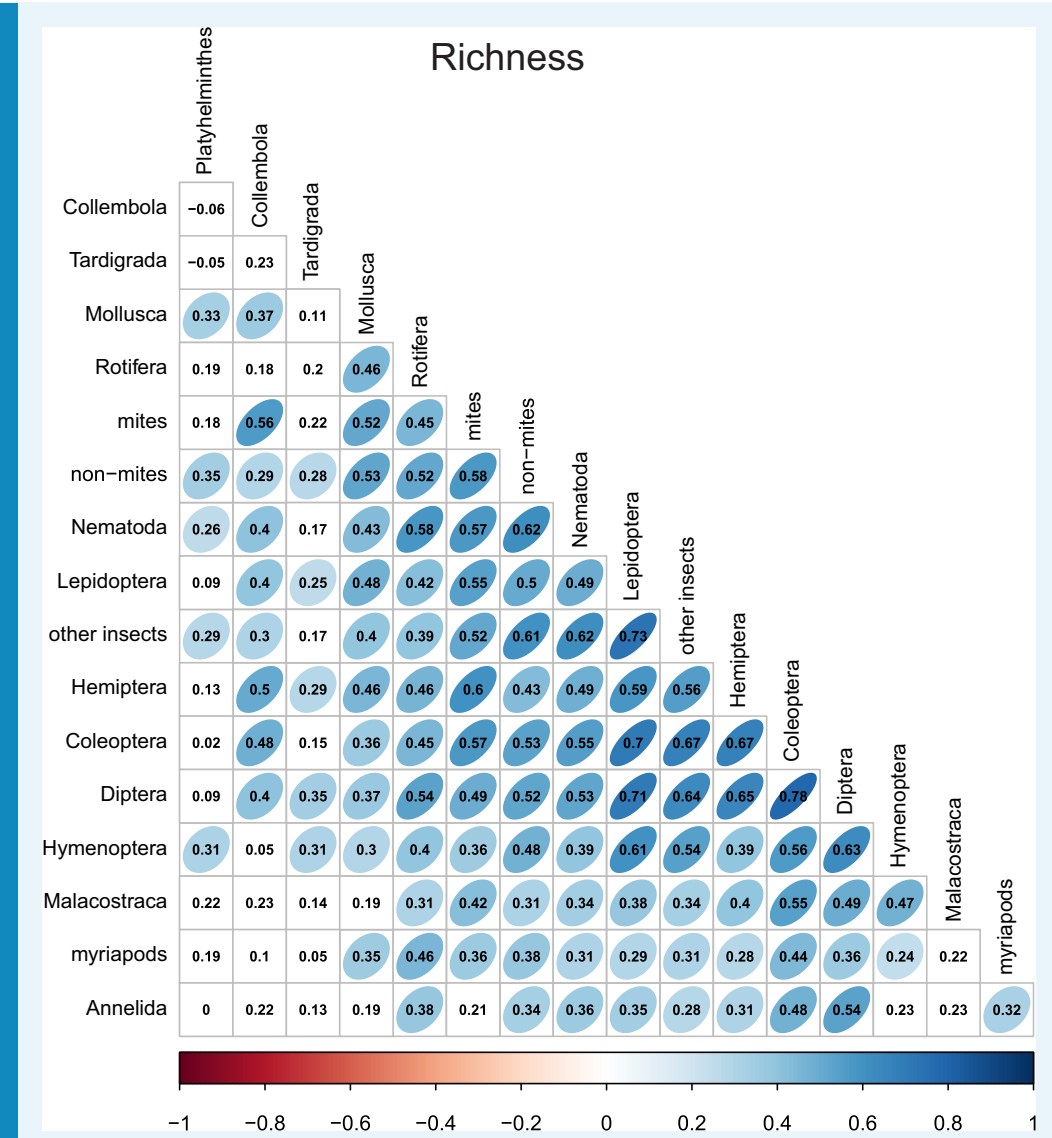

**Appendix 1—figure 7.** Richness correlations between different taxonomic groups. Numbers indicate Pearson correlation coefficients. Ellipse shape and colour represent the magnitude of correlations with p-values≤0.05.

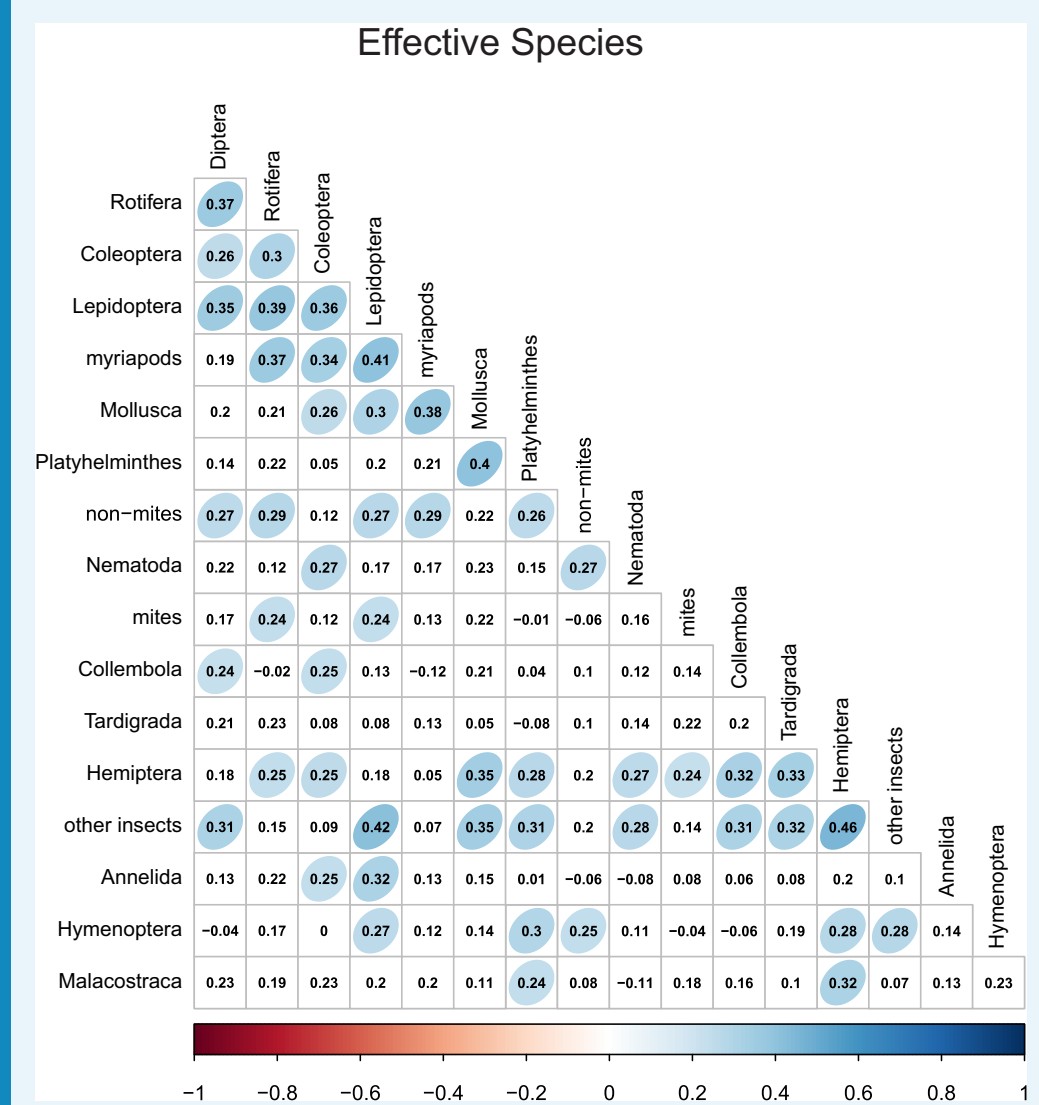

**Appendix 1—figure 8.** Effective species number correlations between different taxonomic groups. Numbers indicate Pearson correlation coefficients. Ellipse shape and colour represent the magnitude of correlations with p-values≤0.05.

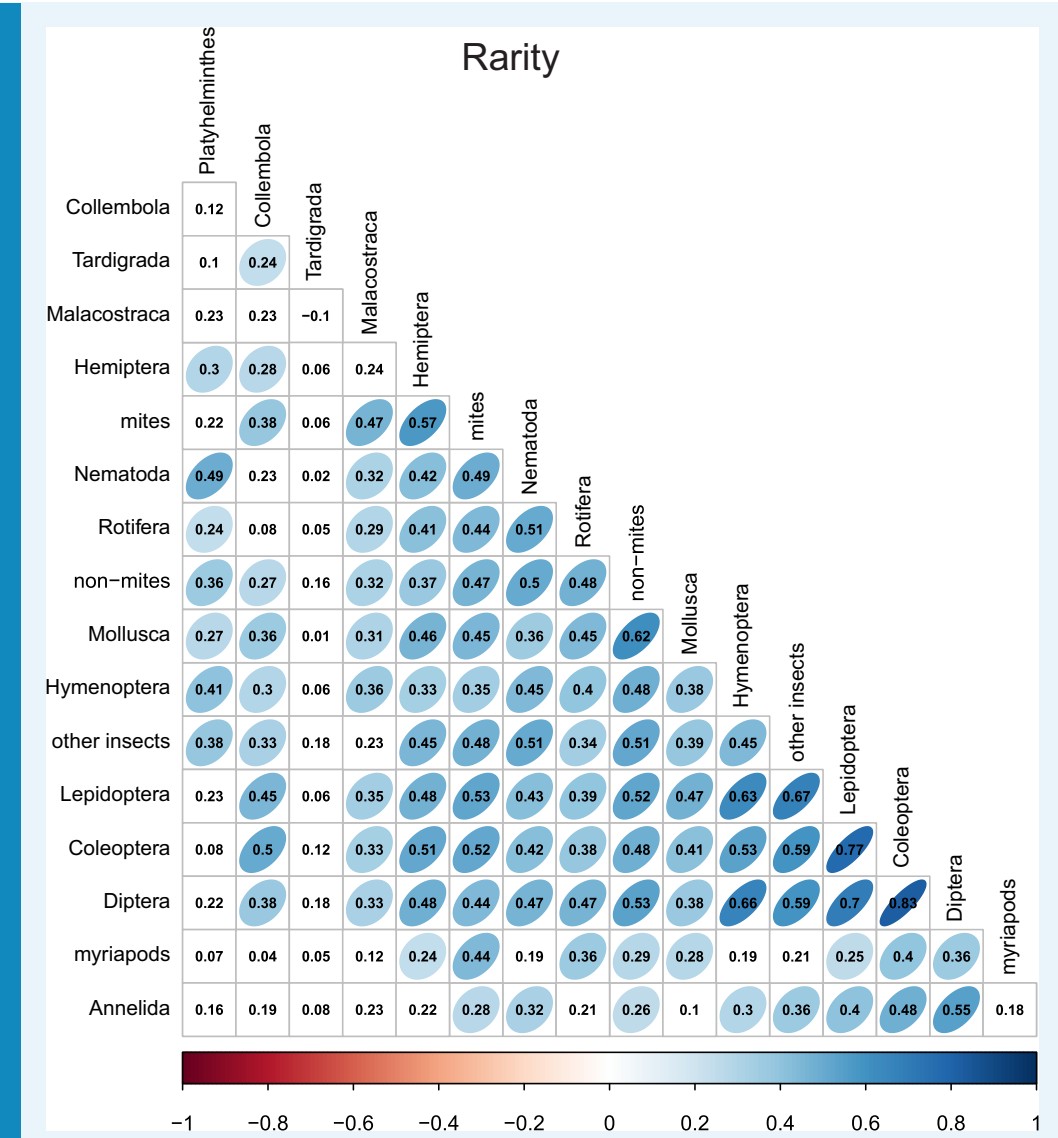

**Appendix 1—figure 9.** Rarity correlations between different taxonomic groups. Numbers indicate Pearson correlation coefficients. Ellipse shape and colour represent the magnitude of correlations with p-values≤0.05.

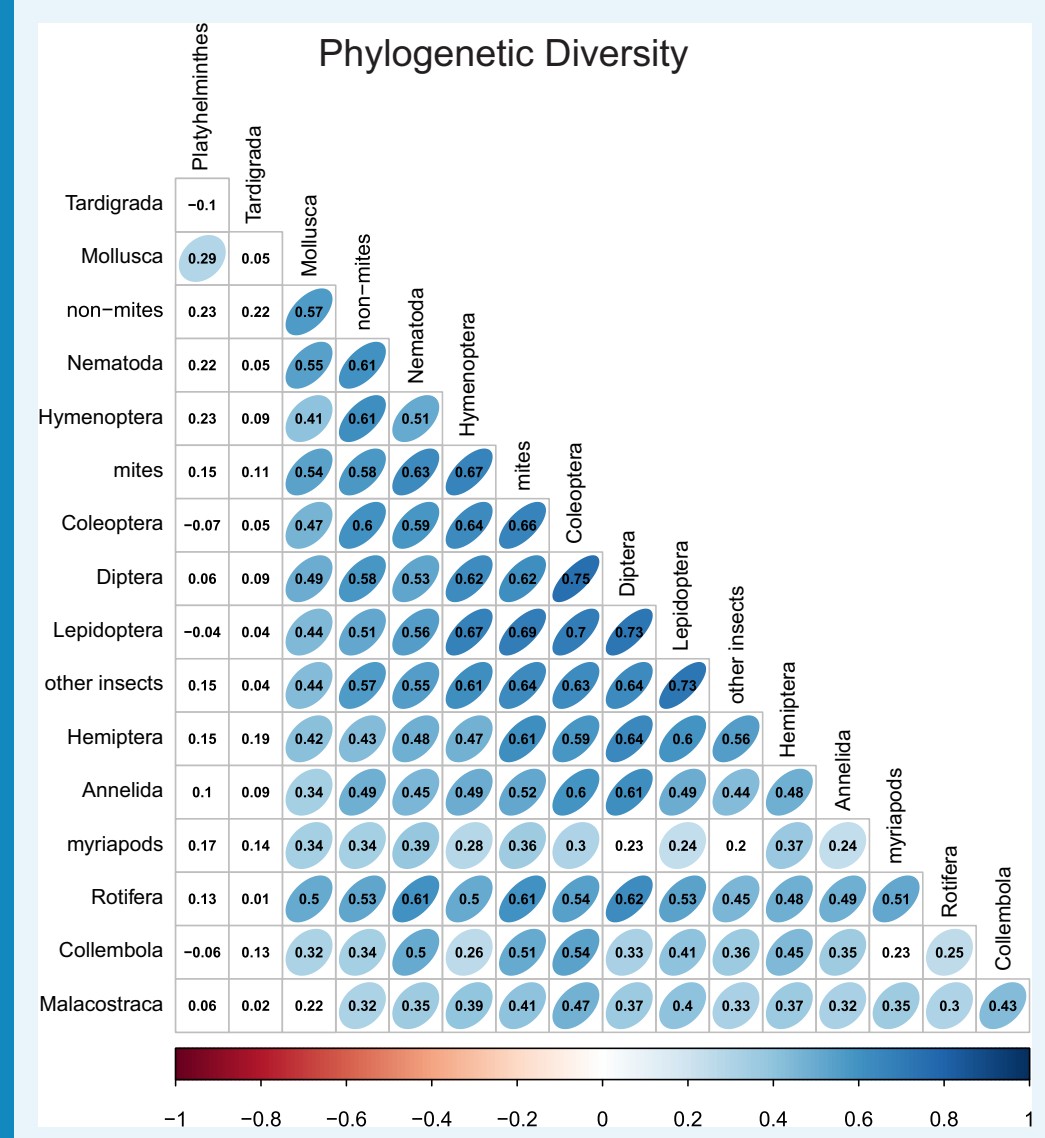

**Appendix 1—figure 10.** Phylogenetic diversity correlations between different taxonomic groups. Numbers indicate Pearson correlation coefficients. Ellipse shape and colour represent the magnitude of correlations with p-values≤0.05.

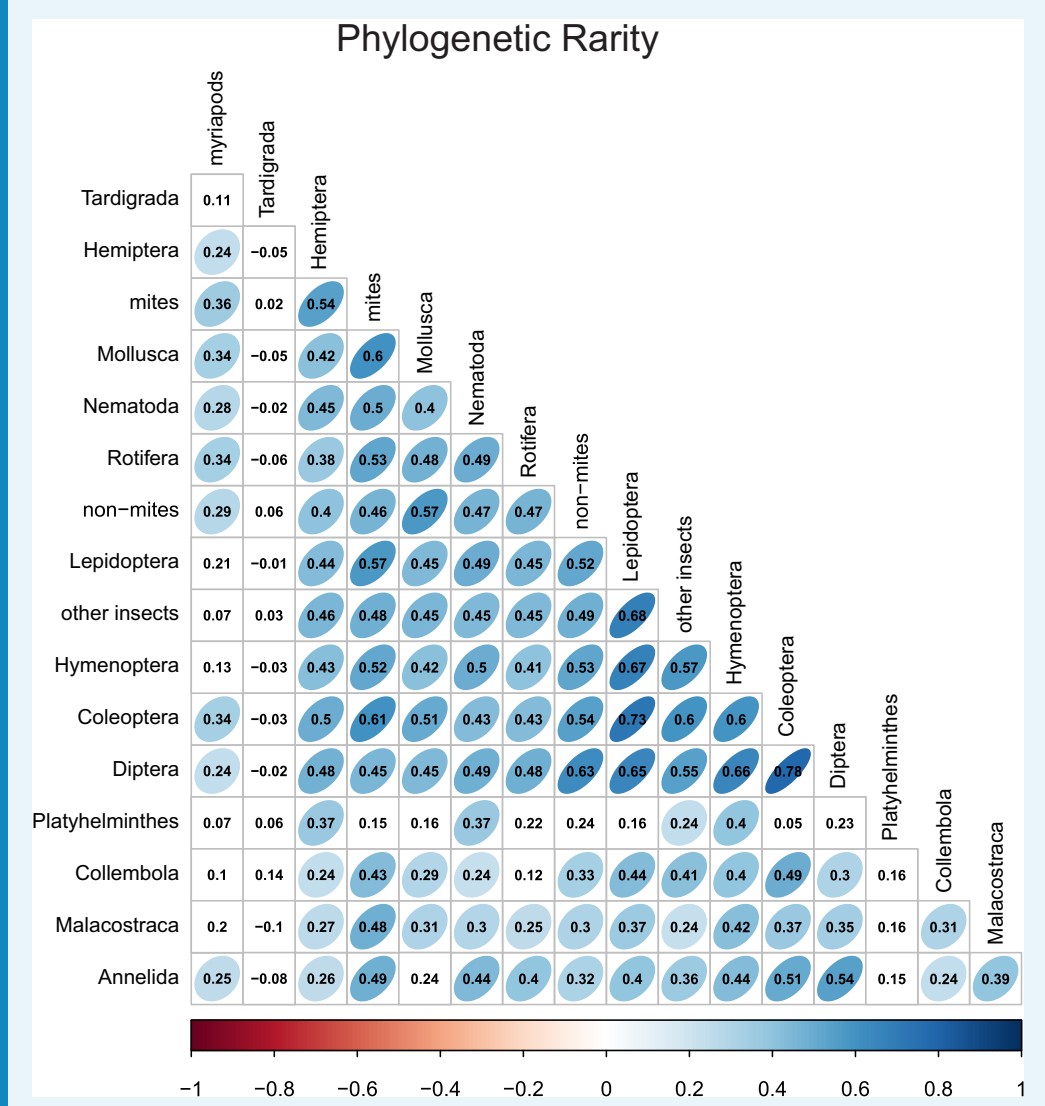

**Appendix 1—figure 11.** Phylogenetic rarity correlations between different taxonomic groups. Numbers indicate Pearson correlation coefficients. Ellipse shape and colour represent the magnitude of correlations with p-values≤0.05.

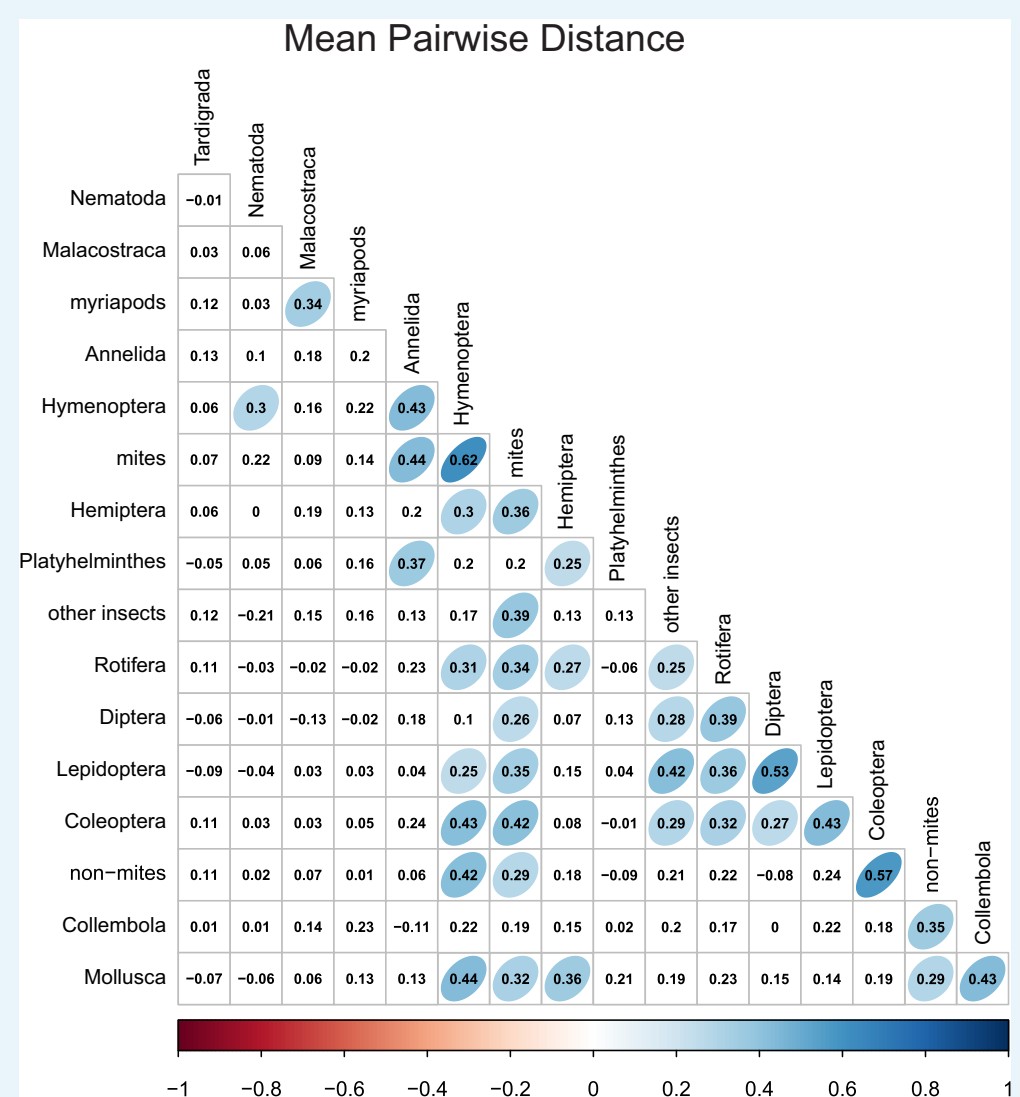

**Appendix 1—figure 12.** Mean pairwise distance correlations between different taxonomic groups. Numbers indicate Pearson correlation coefficients. Ellipse shape and colour represent the magnitude of correlations with p-values≤0.05.

**Appendix 1—table 1.** Results of ANOVA tests for significant derived land-use rank (DLUR) trends for overall soil invertebrate communities and each biodiversity metric.

| Metric | Term | Df | Sum Sq. | Mean Sq. | F stat. | R² | P |
|---|---|---|---|---|---|---|---|
| Richness | DLUR | 1 | 33428.07 | 33428.07 | 8.18 | 0.11 | 0.006 |
| | Residuals | 67 | 273678.56 | 4084.75 | | 0.89 | |
| Effective Species | DLUR | 1 | 1163.40 | 1163.40 | 4.66 | 0.07 | 0.034 |
| | Residuals | 67 | 16728.67 | 249.68 | | 0.93 | |
| Rarity | DLUR | 1 | 25771.74 | 25771.74 | 31.94 | 0.32 | <0.001 |
| | Residuals | 67 | 54061.04 | 806.88 | | 0.68 | |
| Phylogenetic Diversity | DLUR | 1 | 1234.54 | 1234.54 | 11.17 | 0.14 | 0.001 |
| | Residuals | 67 | 7404.70 | 110.52 | | 0.86 | |

*Appendix 1—table 1 continued on next page*

*Appendix 1—table 1 continued*

| Metric | Term | Df | Sum Sq. | Mean Sq. | F stat. | R² | P |
|---|---|---|---|---|---|---|---|
| Phylogenetic Rarity | DLUR | 1 | 311.22 | 311.22 | 31.71 | 0.32 | <0.001 |
| | Residuals | 67 | 657.64 | 9.82 | | 0.68 | |
| Mean Pairwise Distance | DLUR | 1 | 0.00 | 0.00 | 0.99 | 0.01 | 0.324 |
| | Residuals | 67 | 0.11 | 0.00 | | 0.99 | |
| Phylogenetic Diversity SES | DLUR | 1 | 25.63 | 25.63 | 5.62 | 0.08 | 0.021 |
| | Residuals | 67 | 305.46 | 4.56 | | 0.92 | |
| Phylogenetic Rarity SES | DLUR | 1 | 549.79 | 549.79 | 48.95 | 0.42 | <0.001 |
| | Residuals | 67 | 752.56 | 11.23 | | 0.58 | |
| Mean Pairwise Distance SES | DLUR | 1 | 11.16 | 11.16 | 3.28 | 0.05 | 0.075 |
| | Residuals | 67 | 228.00 | 3.40 | | 0.95 | |

**Appendix 1—table 2.** Results of mixed-model ANOVA tests for derived land-use rank (DLUR), land-use category (LCAT), and taxonomic group differences and interactions for each biodiversity metric.

| Metric | Term | Df | Sum sq. | Mean sq. | F stat. | P |
|---|---|---|---|---|---|---|
| Richness | DLUR | 1 | 277.45 | 277.45 | 7.74 | 0.007 |
| | LCAT | 3 | 205.35 | 68.45 | 1.91 | 0.137 |
| | Group | 16 | 15737.52 | 983.60 | 27.43 | <0.001 |
| | DLUR:Group | 16 | 1014.91 | 63.43 | 1.77 | 0.031 |
| | LCAT:Group | 48 | 2425.37 | 50.53 | 1.41 | 0.037 |
| Effective Species | DLUR | 1 | 66.45 | 66.45 | 9.28 | 0.003 |
| | LCAT | 3 | 36.93 | 12.31 | 1.72 | 0.173 |
| | Group | 16 | 3000.93 | 187.56 | 26.19 | <0.001 |
| | DLUR:Group | 16 | 155.71 | 9.73 | 1.36 | 0.155 |
| | LCAT:Group | 48 | 293.70 | 6.12 | 0.85 | 0.749 |
| Rarity | DLUR | 1 | 222.86 | 222.86 | 24.71 | <0.001 |
| | LCAT | 3 | 17.23 | 5.74 | 0.64 | 0.594 |
| | Group | 16 | 3082.62 | 192.66 | 21.36 | <0.001 |
| | DLUR:Group | 16 | 421.79 | 26.36 | 2.92 | <0.001 |
| | LCAT:Group | 48 | 318.82 | 6.64 | 0.74 | 0.908 |
| Phylogenetic Diversity | DLUR | 1 | 12.97 | 12.97 | 12.83 | 0.001 |
| | LCAT | 3 | 4.73 | 1.58 | 1.56 | 0.208 |
| | Group | 16 | 520.03 | 32.50 | 32.14 | <0.001 |
| | DLUR:Group | 16 | 62.75 | 3.92 | 3.88 | <0.001 |
| | LCAT:Group | 48 | 88.35 | 1.84 | 1.82 | <0.001 |
| Phylogenetic Rarity | DLUR | 1 | 4.14 | 4.14 | 31.77 | <0.001 |
| | LCAT | 3 | 0.28 | 0.09 | 0.72 | 0.543 |
| | Group | 16 | 38.74 | 2.42 | 18.56 | <0.001 |
| | DLUR:Group | 16 | 10.12 | 0.63 | 4.85 | <0.001 |
| | LCAT:Group | 48 | 6.94 | 0.14 | 1.11 | 0.288 |
| Mean Pairwise Distance | DLUR | 1 | 0.21 | 0.21 | 2.87 | 0.096 |
| | LCAT | 3 | 0.21 | 0.07 | 0.95 | 0.421 |
| | Group | 16 | 19.11 | 1.19 | 16.40 | <0.001 |

*Appendix 1—table 2 continued on next page*

*Appendix 1—table 2 continued*

| Metric | Term | Df | Sum sq. | Mean sq. | F stat. | P |
|---|---|---|---|---|---|---|
| | DLUR:Group | 16 | 2.93 | 0.18 | 2.51 | 0.001 |
| | LCAT:Group | 48 | 4.20 | 0.09 | 1.20 | 0.169 |

**Appendix 1—table 3.** Results of ANOVA tests for effects of spatial attributes (latitude and altitude) and land-use category on overall invertebrate community biodiversity metrics.

| Metric | Term | Df | Sum Sq. | Mean Sq. | F stat. | $R^2$ | P |
|---|---|---|---|---|---|---|---|
| Richness | Latitude | 1 | 79.49 | 79.49 | 0.02 | 0.000 | 0.882 |
| | Altitude | 1 | 72529.28 | 72529.28 | 20.19 | 0.236 | <0.001 |
| | Land use | 4 | 11761.72 | 2940.43 | 0.82 | 0.038 | 0.518 |
| | Residuals | 62 | 222736.15 | 3592.52 | | 0.725 | |
| Effective Species | Latitude | 1 | 283.47 | 283.47 | 1.19 | 0.016 | 0.280 |
| | Altitude | 1 | 2241.17 | 2241.17 | 9.41 | 0.125 | 0.003 |
| | Land use | 4 | 597.76 | 149.44 | 0.63 | 0.033 | 0.645 |
| | Residuals | 62 | 14769.66 | 238.22 | | 0.825 | |
| Rarity | Latitude | 1 | 465.12 | 465.12 | 0.60 | 0.006 | 0.443 |
| | Altitude | 1 | 17387.74 | 17387.74 | 22.33 | 0.218 | <0.001 |
| | Land use | 4 | 13699.46 | 3424.87 | 4.40 | 0.172 | 0.003 |
| | Residuals | 62 | 48280.46 | 778.72 | | 0.605 | |
| Phylogenetic Diversity | Latitude | 1 | 0.75 | 0.75 | 0.01 | 0.000 | 0.933 |
| | Altitude | 1 | 1740.88 | 1740.88 | 16.78 | 0.202 | <0.001 |
| | Land use | 4 | 464.28 | 116.07 | 1.12 | 0.054 | 0.356 |
| | Residuals | 62 | 6433.33 | 103.76 | | 0.745 | |
| Phylogenetic Rarity | Latitude | 1 | 2.96 | 2.96 | 0.30 | 0.003 | 0.586 |
| | Altitude | 1 | 164.59 | 164.59 | 16.68 | 0.170 | <0.001 |
| | Land use | 4 | 189.57 | 47.39 | 4.80 | 0.196 | 0.002 |
| | Residuals | 62 | 611.74 | 9.87 | | 0.631 | |
| Mean Pairwise Distance | Latitude | 1 | 0.00 | 0.00 | 0.39 | 0.006 | 0.536 |
| | Altitude | 1 | 0.00 | 0.00 | 0.68 | 0.010 | 0.411 |
| | Land use | 4 | 0.01 | 0.00 | 1.41 | 0.082 | 0.241 |
| | Residuals | 62 | 0.10 | 0.00 | | 0.902 | |

**Appendix 1—table 4.** Results of ANOVA tests for effects of the first three components of a PCA on environmental covariates, plus land-use category, on overall invertebrate community biodiversity metrics. A PCA was carried out on spatial (latitude and altitude) and soil chemistry variables (pH, C, N, C:N ratio, Olsen P, Total P, Ca, Mg, K, Na, cation exchange capacity, base saturation), of which the first three components explained 70.25% of variation.

| Metric | Term | Df | Sum Sq. | Mean Sq. | F stat. | $R^2$ | P |
|---|---|---|---|---|---|---|---|
| Richness | PC1 | 1 | 12142.57 | 12142.57 | 3.25 | 0.040 | 0.076 |
| | PC2 | 1 | 26135.36 | 26135.36 | 7.00 | 0.085 | 0.010 |
| | PC3 | 1 | 414.99 | 414.99 | 0.11 | 0.001 | 0.740 |
| | Land use | 4 | 40528.70 | 10132.18 | 2.71 | 0.132 | 0.038 |
| | Residuals | 61 | 227885.01 | 3735.82 | | 0.742 | |
| Effective Species | PC1 | 1 | 497.15 | 497.15 | 2.02 | 0.028 | 0.161 |

*Appendix 1—table 4 continued on next page*

*Appendix 1—table 4 continued*

| Metric | Term | Df | Sum Sq. | Mean Sq. | F stat. | R² | P |
|---|---|---|---|---|---|---|---|
| | PC2 | 1 | 618.84 | 618.84 | 2.51 | 0.035 | 0.118 |
| | PC3 | 1 | 11.19 | 11.19 | 0.05 | 0.001 | 0.832 |
| | Land use | 4 | 1725.81 | 431.45 | 1.75 | 0.096 | 0.151 |
| | Residuals | 61 | 15039.07 | 246.54 | | 0.841 | |
| Rarity | PC1 | 1 | 11905.88 | 11905.88 | 15.25 | 0.149 | <0.001 |
| | PC2 | 1 | 7487.41 | 7487.41 | 9.59 | 0.094 | 0.003 |
| | PC3 | 1 | 233.36 | 233.36 | 0.30 | 0.003 | 0.587 |
| | Land use | 4 | 12569.11 | 3142.28 | 4.02 | 0.157 | 0.006 |
| | Residuals | 61 | 47637.01 | 780.93 | | 0.597 | |
| Phylogenetic Diversity | PC1 | 1 | 503.99 | 503.99 | 4.79 | 0.058 | 0.032 |
| | PC2 | 1 | 812.16 | 812.16 | 7.72 | 0.094 | 0.007 |
| | PC3 | 1 | 10.98 | 10.98 | 0.10 | 0.001 | 0.748 |
| | Land use | 4 | 897.84 | 224.46 | 2.13 | 0.104 | 0.087 |
| | Residuals | 61 | 6414.27 | 105.15 | | 0.742 | |
| Phylogenetic Rarity | PC1 | 1 | 147.45 | 147.45 | 15.35 | 0.152 | <0.001 |
| | PC2 | 1 | 98.38 | 98.38 | 10.24 | 0.102 | 0.002 |
| | PC3 | 1 | 10.34 | 10.34 | 1.08 | 0.011 | 0.304 |
| | Land use | 4 | 126.54 | 31.63 | 3.29 | 0.131 | 0.017 |
| | Residuals | 61 | 586.15 | 9.61 | | 0.605 | |
| Mean Pairwise Distance | PC1 | 1 | 0.002 | 0.002 | 1.432 | 0.021 | 0.236 |
| | PC2 | 1 | 0.001 | 0.001 | 0.478 | 0.007 | 0.492 |
| | PC3 | 1 | 0.002 | 0.002 | 0.996 | 0.015 | 0.322 |
| | Land use | 4 | 0.006 | 0.001 | 0.863 | 0.051 | 0.491 |
| | Residuals | 61 | 0.105 | 0.002 | | 0.906 | |

**Appendix 1—table 5.** Results of ANOVA tests for significant derived land-use rank (DLUR) trends for each taxonomic group and biodiversity metric.

| Metric | Group | Term | Df | Sum Sq. | Mean Sq. | F stat. | R² | P |
|---|---|---|---|---|---|---|---|---|
| Richness | Collembola | DLUR | 1 | 28.935 | 28.935 | 2.649 | 0.039 | 0.108 |
| | | Residuals | 65 | 710.110 | 10.925 | | 0.961 | |
| | Coleoptera | DLUR | 1 | 335.849 | 335.849 | 9.961 | 0.131 | 0.002 |
| | | Residuals | 66 | 2225.210 | 33.715 | | 0.869 | |
| | Diptera | DLUR | 1 | 615.926 | 615.926 | 18.012 | 0.214 | <0.001 |
| | | Residuals | 66 | 2256.839 | 34.195 | | 0.786 | |
| | Hymenoptera | DLUR | 1 | 293.041 | 293.041 | 19.024 | 0.229 | <0.001 |
| | | Residuals | 64 | 985.823 | 15.403 | | 0.771 | |
| | Lepidoptera | DLUR | 1 | 476.785 | 476.785 | 19.328 | 0.227 | <0.001 |
| | | Residuals | 66 | 1628.083 | 24.668 | | 0.773 | |
| | Hemiptera | DLUR | 1 | 26.788 | 26.788 | 2.920 | 0.044 | 0.092 |
| | | Residuals | 64 | 587.166 | 9.174 | | 0.956 | |
| | other insects | DLUR | 1 | 74.531 | 74.531 | 8.968 | 0.123 | 0.004 |
| | | Residuals | 64 | 531.909 | 8.311 | | 0.877 | |

*Appendix 1—table 5 continued on next page*

Appendix 1—table 5 continued

| Metric | Group | Term | Df | Sum Sq. | Mean Sq. | F stat. | R² | P |
|---|---|---|---|---|---|---|---|---|
| | non-mites | DLUR | 1 | 45.412 | 45.412 | 2.924 | 0.043 | 0.092 |
| | | Residuals | 65 | 1009.454 | 15.530 | | 0.957 | |
| | mites | DLUR | 1 | 0.281 | 0.281 | 0.008 | 0.000 | 0.930 |
| | | Residuals | 66 | 2359.719 | 35.753 | | 1.000 | |
| | Malacostraca | DLUR | 1 | 0.044 | 0.044 | 0.046 | 0.001 | 0.831 |
| | | Residuals | 34 | 32.706 | 0.962 | | 0.999 | |
| | myriapods | DLUR | 1 | 1.518 | 1.518 | 0.555 | 0.017 | 0.462 |
| | | Residuals | 32 | 87.453 | 2.733 | | 0.983 | |
| | Annelida | DLUR | 1 | 67.637 | 67.637 | 4.456 | 0.065 | 0.039 |
| | | Residuals | 64 | 971.393 | 15.178 | | 0.935 | |
| | Mollusca | DLUR | 1 | 0.240 | 0.240 | 0.014 | 0.000 | 0.906 |
| | | Residuals | 64 | 1082.245 | 16.910 | | 1.000 | |
| | Nematoda | DLUR | 1 | 110.491 | 110.491 | 0.559 | 0.008 | 0.457 |
| | | Residuals | 67 | 13243.798 | 197.669 | | 0.992 | |
| | Platyhelminthes | DLUR | 1 | 0.175 | 0.175 | 0.195 | 0.004 | 0.661 |
| | | Residuals | 49 | 43.982 | 0.898 | | 0.996 | |
| | Rotifera | DLUR | 1 | 201.555 | 201.555 | 0.635 | 0.010 | 0.428 |
| | | Residuals | 66 | 20954.136 | 317.487 | | 0.990 | |
| | Tardigrada | DLUR | 1 | 0.177 | 0.177 | 0.469 | 0.015 | 0.499 |
| | | Residuals | 30 | 11.323 | 0.377 | | 0.985 | |
| Effective Species | Collembola | DLUR | 1 | 0.530 | 0.530 | 0.273 | 0.004 | 0.603 |
| | | Residuals | 65 | 126.200 | 1.942 | | 0.996 | |
| | Coleoptera | DLUR | 1 | 13.595 | 13.595 | 3.866 | 0.055 | 0.053 |
| | | Residuals | 66 | 232.067 | 3.516 | | 0.945 | |
| | Diptera | DLUR | 1 | 44.602 | 44.602 | 10.967 | 0.142 | 0.002 |
| | | Residuals | 66 | 268.411 | 4.067 | | 0.858 | |
| | Hymenoptera | DLUR | 1 | 21.826 | 21.826 | 4.614 | 0.067 | 0.036 |
| | | Residuals | 64 | 302.768 | 4.731 | | 0.933 | |
| | Lepidoptera | DLUR | 1 | 62.719 | 62.719 | 12.384 | 0.158 | 0.001 |
| | | Residuals | 66 | 334.257 | 5.064 | | 0.842 | |
| | Hemiptera | DLUR | 1 | 0.666 | 0.666 | 0.281 | 0.004 | 0.598 |
| | | Residuals | 64 | 151.775 | 2.371 | | 0.996 | |
| | other insects | DLUR | 1 | 1.186 | 1.186 | 0.849 | 0.013 | 0.360 |
| | | Residuals | 64 | 89.462 | 1.398 | | 0.987 | |
| | non-mites | DLUR | 1 | 6.539 | 6.539 | 1.997 | 0.030 | 0.162 |
| | | Residuals | 65 | 212.805 | 3.274 | | 0.970 | |
| | mites | DLUR | 1 | 1.064 | 1.064 | 0.256 | 0.004 | 0.615 |
| | | Residuals | 66 | 274.716 | 4.162 | | 0.996 | |
| | Malacostraca | DLUR | 1 | 0.007 | 0.007 | 0.036 | 0.001 | 0.852 |
| | | Residuals | 34 | 6.754 | 0.199 | | 0.999 | |
| | myriapods | DLUR | 1 | 0.637 | 0.637 | 0.610 | 0.019 | 0.441 |
| | | Residuals | 32 | 33.414 | 1.044 | | 0.981 | |
| | Annelida | DLUR | 1 | 15.530 | 15.530 | 9.802 | 0.133 | 0.003 |

*Appendix 1—table 5 continued on next page*

*Appendix 1—table 5 continued*

| Metric | Group | Term | Df | Sum Sq. | Mean Sq. | F stat. | $R^2$ | P |
|---|---|---|---|---|---|---|---|---|
| | | Residuals | 64 | 101.400 | 1.584 | | 0.867 | |
| | Mollusca | DLUR | 1 | 4.329 | 4.329 | 1.021 | 0.016 | 0.316 |
| | | Residuals | 64 | 271.445 | 4.241 | | 0.984 | |
| | Nematoda | DLUR | 1 | 16.682 | 16.682 | 0.571 | 0.008 | 0.452 |
| | | Residuals | 67 | 1955.701 | 29.190 | | 0.992 | |
| | Platyhelminthes | DLUR | 1 | 0.094 | 0.094 | 0.340 | 0.007 | 0.563 |
| | | Residuals | 49 | 13.622 | 0.278 | | 0.993 | |
| | Rotifera | DLUR | 1 | 131.612 | 131.612 | 2.615 | 0.038 | 0.111 |
| | | Residuals | 66 | 3321.550 | 50.327 | | 0.962 | |
| | Tardigrada | DLUR | 1 | 0.174 | 0.174 | 0.630 | 0.021 | 0.434 |
| | | Residuals | 30 | 8.291 | 0.276 | | 0.979 | |
| Rarity | Collembola | DLUR | 1 | 2.891 | 2.891 | 1.331 | 0.020 | 0.253 |
| | | Residuals | 65 | 141.173 | 2.172 | | 0.980 | |
| | Coleoptera | DLUR | 1 | 263.306 | 263.306 | 25.786 | 0.281 | <0.001 |
| | | Residuals | 66 | 673.930 | 10.211 | | 0.719 | |
| | Diptera | DLUR | 1 | 422.250 | 422.250 | 51.691 | 0.439 | <0.001 |
| | | Residuals | 66 | 539.139 | 8.169 | | 0.561 | |
| | Hymenoptera | DLUR | 1 | 102.088 | 102.088 | 19.399 | 0.233 | <0.001 |
| | | Residuals | 64 | 336.809 | 5.263 | | 0.767 | |
| | Lepidoptera | DLUR | 1 | 256.685 | 256.685 | 36.200 | 0.354 | <0.001 |
| | | Residuals | 66 | 467.983 | 7.091 | | 0.646 | |
| | Hemiptera | DLUR | 1 | 16.816 | 16.816 | 7.048 | 0.099 | 0.010 |
| | | Residuals | 64 | 152.695 | 2.386 | | 0.901 | |
| | other insects | DLUR | 1 | 41.828 | 41.828 | 14.903 | 0.189 | <0.001 |
| | | Residuals | 64 | 179.631 | 2.807 | | 0.811 | |
| | non-mites | DLUR | 1 | 65.614 | 65.614 | 13.757 | 0.175 | <0.001 |
| | | Residuals | 65 | 310.020 | 4.770 | | 0.825 | |
| | mites | DLUR | 1 | 37.895 | 37.895 | 5.675 | 0.079 | 0.020 |
| | | Residuals | 66 | 440.698 | 6.677 | | 0.921 | |
| | Malacostraca | DLUR | 1 | 0.126 | 0.126 | 0.241 | 0.007 | 0.627 |
| | | Residuals | 34 | 17.745 | 0.522 | | 0.993 | |
| | myriapods | DLUR | 1 | 4.426 | 4.426 | 3.511 | 0.099 | 0.070 |
| | | Residuals | 32 | 40.347 | 1.261 | | 0.901 | |
| | Annelida | DLUR | 1 | 47.966 | 47.966 | 17.057 | 0.210 | <0.001 |
| | | Residuals | 64 | 179.972 | 2.812 | | 0.790 | |
| | Mollusca | DLUR | 1 | 15.081 | 15.081 | 2.992 | 0.045 | 0.088 |
| | | Residuals | 64 | 322.535 | 5.040 | | 0.955 | |
| | Nematoda | DLUR | 1 | 197.691 | 197.691 | 6.008 | 0.082 | 0.017 |
| | | Residuals | 67 | 2204.664 | 32.905 | | 0.918 | |
| | Platyhelminthes | DLUR | 1 | 0.333 | 0.333 | 0.854 | 0.017 | 0.360 |
| | | Residuals | 49 | 19.112 | 0.390 | | 0.983 | |
| | Rotifera | DLUR | 1 | 157.017 | 157.017 | 2.033 | 0.030 | 0.159 |
| | | Residuals | 66 | 5097.292 | 77.232 | | 0.970 | |

| Metric | Group | Term | Df | Sum Sq. | Mean Sq. | F stat. | R² | P |
|---|---|---|---|---|---|---|---|---|
| | Tardigrada | DLUR | 1 | 0.304 | 0.304 | 1.197 | 0.038 | 0.283 |
| | | Residuals | 30 | 7.613 | 0.254 | | 0.962 | |
| Phylogenetic Diversity | Collembola | DLUR | 1 | 0.711 | 0.711 | 1.505 | 0.023 | 0.224 |
| | | Residuals | 65 | 30.718 | 0.473 | | 0.977 | |
| | Coleoptera | DLUR | 1 | 21.952 | 21.952 | 9.688 | 0.128 | 0.003 |
| | | Residuals | 66 | 149.544 | 2.266 | | 0.872 | |
| | Diptera | DLUR | 1 | 40.106 | 40.106 | 19.384 | 0.227 | <0.001 |
| | | Residuals | 66 | 136.557 | 2.069 | | 0.773 | |
| | Hymenoptera | DLUR | 1 | 21.848 | 21.848 | 12.731 | 0.166 | 0.001 |
| | | Residuals | 64 | 109.830 | 1.716 | | 0.834 | |
| | Lepidoptera | DLUR | 1 | 42.880 | 42.880 | 20.442 | 0.236 | <0.001 |
| | | Residuals | 66 | 138.446 | 2.098 | | 0.764 | |
| | Hemiptera | DLUR | 1 | 5.773 | 5.773 | 3.883 | 0.057 | 0.053 |
| | | Residuals | 64 | 95.147 | 1.487 | | 0.943 | |
| | other insects | DLUR | 1 | 18.456 | 18.456 | 13.631 | 0.176 | <0.001 |
| | | Residuals | 64 | 86.652 | 1.354 | | 0.824 | |
| | non-mites | DLUR | 1 | 13.612 | 13.612 | 7.064 | 0.098 | 0.010 |
| | | Residuals | 65 | 125.259 | 1.927 | | 0.902 | |
| | mites | DLUR | 1 | 7.975 | 7.975 | 2.926 | 0.042 | 0.092 |
| | | Residuals | 66 | 179.924 | 2.726 | | 0.958 | |
| | Malacostraca | DLUR | 1 | 0.468 | 0.468 | 0.755 | 0.022 | 0.391 |
| | | Residuals | 34 | 21.069 | 0.620 | | 0.978 | |
| | myriapods | DLUR | 1 | 0.002 | 0.002 | 0.005 | 0.000 | 0.944 |
| | | Residuals | 32 | 15.613 | 0.488 | | 1.000 | |
| | Annelida | DLUR | 1 | 10.571 | 10.571 | 13.578 | 0.175 | <0.001 |
| | | Residuals | 64 | 49.825 | 0.779 | | 0.825 | |
| | Mollusca | DLUR | 1 | 1.072 | 1.072 | 0.379 | 0.006 | 0.540 |
| | | Residuals | 64 | 181.156 | 2.831 | | 0.994 | |
| | Nematoda | DLUR | 1 | 0.002 | 0.002 | 0.000 | 0.000 | 0.985 |
| | | Residuals | 67 | 305.660 | 4.562 | | 1.000 | |
| | Platyhelminthes | DLUR | 1 | 1.314 | 1.314 | 2.057 | 0.040 | 0.158 |
| | | Residuals | 49 | 31.305 | 0.639 | | 0.960 | |
| | Rotifera | DLUR | 1 | 5.444 | 5.444 | 2.665 | 0.039 | 0.107 |
| | | Residuals | 66 | 134.809 | 2.043 | | 0.961 | |
| | Tardigrada | DLUR | 1 | 0.174 | 0.174 | 1.594 | 0.050 | 0.217 |
| | | Residuals | 30 | 3.279 | 0.109 | | 0.950 | |
| Phylogenetic Rarity | Collembola | DLUR | 1 | 0.165 | 0.165 | 4.070 | 0.059 | 0.048 |
| | | Residuals | 65 | 2.640 | 0.041 | | 0.941 | |
| | Coleoptera | DLUR | 1 | 8.151 | 8.151 | 26.035 | 0.283 | <0.001 |
| | | Residuals | 66 | 20.663 | 0.313 | | 0.717 | |
| | Diptera | DLUR | 1 | 10.221 | 10.221 | 40.013 | 0.377 | <0.001 |
| | | Residuals | 66 | 16.859 | 0.255 | | 0.623 | |

*Appendix 1—table 5 continued*

| Metric | Group | Term | Df | Sum Sq. | Mean Sq. | F stat. | R² | P |
|---|---|---|---|---|---|---|---|---|
| | Hymenoptera | DLUR | 1 | 3.536 | 3.536 | 19.319 | 0.232 | <0.001 |
| | | Residuals | 64 | 11.713 | 0.183 | | 0.768 | |
| | Lepidoptera | DLUR | 1 | 6.931 | 6.931 | 36.877 | 0.358 | <0.001 |
| | | Residuals | 66 | 12.405 | 0.188 | | 0.642 | |
| | Hemiptera | DLUR | 1 | 0.467 | 0.467 | 4.095 | 0.060 | 0.047 |
| | | Residuals | 64 | 7.306 | 0.114 | | 0.940 | |
| | other insects | DLUR | 1 | 2.752 | 2.752 | 15.755 | 0.198 | <0.001 |
| | | Residuals | 64 | 11.180 | 0.175 | | 0.802 | |
| | non-mites | DLUR | 1 | 4.073 | 4.073 | 16.159 | 0.199 | <0.001 |
| | | Residuals | 65 | 16.384 | 0.252 | | 0.801 | |
| | mites | DLUR | 1 | 2.303 | 2.303 | 11.052 | 0.143 | 0.001 |
| | | Residuals | 66 | 13.752 | 0.208 | | 0.857 | |
| | Malacostraca | DLUR | 1 | 0.021 | 0.021 | 0.314 | 0.009 | 0.579 |
| | | Residuals | 34 | 2.236 | 0.066 | | 0.991 | |
| | myriapods | DLUR | 1 | 0.200 | 0.200 | 4.708 | 0.128 | 0.038 |
| | | Residuals | 32 | 1.359 | 0.042 | | 0.872 | |
| | Annelida | DLUR | 1 | 2.024 | 2.024 | 25.966 | 0.289 | <0.001 |
| | | Residuals | 64 | 4.989 | 0.078 | | 0.711 | |
| | Mollusca | DLUR | 1 | 1.418 | 1.418 | 3.570 | 0.053 | 0.063 |
| | | Residuals | 64 | 25.424 | 0.397 | | 0.947 | |
| | Nematoda | DLUR | 1 | 1.574 | 1.574 | 4.943 | 0.069 | 0.030 |
| | | Residuals | 67 | 21.328 | 0.318 | | 0.931 | |
| | Platyhelminthes | DLUR | 1 | 0.000 | 0.000 | 0.002 | 0.000 | 0.966 |
| | | Residuals | 49 | 2.873 | 0.059 | | 1.000 | |
| | Rotifera | DLUR | 1 | 0.701 | 0.701 | 3.020 | 0.044 | 0.087 |
| | | Residuals | 66 | 15.321 | 0.232 | | 0.956 | |
| | Tardigrada | DLUR | 1 | 0.095 | 0.095 | 3.543 | 0.106 | 0.070 |
| | | Residuals | 30 | 0.801 | 0.027 | | 0.894 | |
| Mean Pairwise Distance | Collembola | DLUR | 1 | 0.028 | 0.028 | 1.193 | 0.018 | 0.279 |
| | | Residuals | 65 | 1.502 | 0.023 | | 0.982 | |
| | Coleoptera | DLUR | 1 | 0.002 | 0.002 | 0.097 | 0.001 | 0.757 |
| | | Residuals | 66 | 1.338 | 0.020 | | 0.999 | |
| | Diptera | DLUR | 1 | 0.005 | 0.005 | 0.127 | 0.002 | 0.722 |
| | | Residuals | 66 | 2.375 | 0.036 | | 0.998 | |
| | Hymenoptera | DLUR | 1 | 0.421 | 0.421 | 7.001 | 0.099 | 0.010 |
| | | Residuals | 64 | 3.850 | 0.060 | | 0.901 | |
| | Lepidoptera | DLUR | 1 | 0.002 | 0.002 | 0.064 | 0.001 | 0.801 |
| | | Residuals | 66 | 2.325 | 0.035 | | 0.999 | |
| | Hemiptera | DLUR | 1 | 0.113 | 0.113 | 1.211 | 0.019 | 0.275 |
| | | Residuals | 64 | 5.968 | 0.093 | | 0.981 | |
| | other insects | DLUR | 1 | 0.039 | 0.039 | 0.631 | 0.010 | 0.430 |
| | | Residuals | 64 | 3.986 | 0.062 | | 0.990 | |

*Appendix 1—table 5 continued on next page*

*Appendix 1—table 5 continued*

| Metric | Group | Term | Df | Sum Sq. | Mean Sq. | F stat. | $R^2$ | P |
|---|---|---|---|---|---|---|---|---|
| | non-mites | DLUR | 1 | 0.211 | 0.211 | 2.825 | 0.042 | 0.098 |
| | | Residuals | 65 | 4.847 | 0.075 | | 0.958 | |
| | Mites | DLUR | 1 | 0.467 | 0.467 | 13.885 | 0.174 | <0.001 |
| | | Residuals | 66 | 2.222 | 0.034 | | 0.826 | |
| | Malacostraca | DLUR | 1 | 0.689 | 0.689 | 1.957 | 0.054 | 0.171 |
| | | Residuals | 34 | 11.968 | 0.352 | | 0.946 | |
| | myriapods | DLUR | 1 | 0.137 | 0.137 | 0.598 | 0.018 | 0.445 |
| | | Residuals | 32 | 7.318 | 0.229 | | 0.982 | |
| | Annelida | DLUR | 1 | 0.436 | 0.436 | 7.805 | 0.109 | 0.007 |
| | | Residuals | 64 | 3.574 | 0.056 | | 0.891 | |
| | Mollusca | DLUR | 1 | 0.228 | 0.228 | 2.577 | 0.039 | 0.113 |
| | | Residuals | 64 | 5.651 | 0.088 | | 0.961 | |
| | Nematoda | DLUR | 1 | 0.001 | 0.001 | 0.168 | 0.003 | 0.683 |
| | | Residuals | 67 | 0.551 | 0.008 | | 0.997 | |
| | Platyhelminthes | DLUR | 1 | 0.540 | 0.540 | 1.528 | 0.030 | 0.222 |
| | | Residuals | 49 | 17.313 | 0.353 | | 0.970 | |
| | Rotifera | DLUR | 1 | 0.007 | 0.007 | 14.178 | 0.177 | <0.001 |
| | | Residuals | 66 | 0.033 | 0.001 | | 0.823 | |
| | Tardigrada | DLUR | 1 | 0.134 | 0.134 | 1.377 | 0.044 | 0.250 |
| | | Residuals | 30 | 2.910 | 0.097 | | 0.956 | |

