## [Decision Letter]

**Acceptance summary:**

Five categories of land-use that could be ranked according to increasing human impact showed strong linear declines in the majority of 17 invertebrate soil animal groups. These negative impacts were best detected when phylogenetic rarity was used as a measure whereas other diversity measures were less responsive and may therefore be less useful indicators. The comprehensive assessment of the soil invertebrate communities was made possible by the use of DNA metabarcoding.

**Decision letter after peer review:**

Thank you for submitting your article "Endemism is a better indicator of soil invertebrate biodiversity loss with land use change than richness or diversity" for consideration by *eLife*. Your article has been reviewed by three peer reviewers, including Bernhard Schmid as the Reviewing Editor and Reviewer #1, and the evaluation has been overseen by Ian Baldwin as the Senior Editor. The following individual involved in review of your submission has agreed to reveal their identity: Marc Cadotte (Reviewer #3).

The reviewers have discussed the reviews with one another and the Reviewing Editor has drafted this decision to help you prepare a revised submission.

Summary:

This manuscript represents a major effort to use DNA barcoding in analyzing soil communities at 75 sites across New Zealand. More than 11'000 taxa could be identified and assigned to different groups of invertebrates and microbes. The hypothesis is that diversity decreases along a gradient of land-use intensity represented by five land-use categories natural forest -> planted forest -> low-producing grassland -> high-producing grassland -> perennial cropland. This hypothesis is confirmed, especially if species are weighted by the reciprocal of the number of sites at which they occur, suggesting that they are habitat specialists and therefore respond most strongly to land use.

Essential revisions:

The manuscript should be improved in three respects:

First, there are several terms that are not well defined in the manuscript. "Land use" is sometimes considered as a state of an ecosystem and then called land-use category (please always use the hyphen when "land-use" is an adjectival noun) but sometimes considered as land-use change or intensification etc. Please use consistent terms throughout and make sure you define them well. In this context and related to the second main point I suggest that you put the five land-use categories explicitly into a linear sequence from one to five and call it land-use intensity.

Endemism is defined in the Materials and methods, but all reviewers would prefer a more generic term such as rare species because you cannot extrapolate from 75 sites to the entire area of your country, which would be necessary to know if a species that only occurred at 1 or 2 sites was geographically restricted. You should also mention the caveat that some of your rare species could just be "sink" or "passenger" species not adapted to the site conditions, while your hypothesis assumes that they are habitat specialists. One reviewer suggests in this context that you could actually do a sensitivity analysis without the rarest species that only occur at one site.

Second, you should get much more from your data if you expand your analyses. In principle, you selected your five categories such that they can be ordered along a gradient. Best would be if you could calculate some value for land-use intensity for each category or even each site, but even in the absence of this you are allowed to simply assign land-use intensity values from 1-5 and use this linear contrast in all ANOVA (the easiest way is to code land-use categories from 1-5 and use it as continuous variable LUI and factor LUC: then fit LUI+LUC in this order and you get a stronger test of your main hypothesis, namely the linear contrast LUI, and a test for deviations from the linear contrast, which is measured by LUC when fitted after LUI). Please replace all box-plots with means +/- standard errors. In addition, you could show regression lines for the LUI-gradient wherever it is significant. There would be the more sophisticated method of isotonic regression (Gaines, Steven D., and William R. Rice. 1990. Analysis of Biological Data When there are Ordered Expectations. The American Naturalist 135: 310-317.), but clearly the simpler method indicated above is also valid.

The above will remove the need for the excessive use of pairwise comparisons which are statistically not independent and ill-suited to test ordered hypotheses. A further reduction of unnecessary repeats and more comprehensive analysis is to compare the responses of the different taxa to LUI(+LUC) in a single ANOVA by putting the metrics for the different taxa into a single column and add a column with the taxa identities (or even two, one for phyla and one for classes/orders). You can then fit the following ANOVA-model:

Metric ~ LUI+LUC+phylum+order+LUI:phylum+LUC:phylum+LUI:order+LUC:order, or more simply without LUC terms and only using phylum or order. The interactions indicated by ":" test if the different taxonomic groups respond differently. Using the above you can exchange Figure 3—source data 1B and omit Supplementary Table 4. There is still the problem of comparing the R^2^ values (or better just use %SS, which are increments in multiple R^2^) mentioned by one reviewer and we ask you to consider their suggestions. It also seems you analyzed R^2^ values as (meta-)data; this requires explanation and justification in the Materials and methods, because R^2^ values will have a special distribution that deviates from normal.

Furthermore, as noted by one reviewer, from your Figure 6 it appears that the land-use categories are not well "randomized" across space and thus covariates could explain some of your results. You could include some of these first in the fitting sequence of ANOVAs, but we accept that you don't want to put in too many covariates because there are not so many sites to test more complicated models and sometimes correction for covariates can be unjustified if they are not a "true cause" of an observed effect.

One reviewer also suggests that you should try phylogenetic metrics that are not mathematically related to richness.

Your suggestion that the results imply homogenization needs to be tested statistically, which in fact can be done quite easily e.g. by calculating β diversities. You also use a term "heterogeneity", which obviously is taken from output of an R function. However, you cannot expect readers to check this up but rather need to give a clear definition of heterogeneity and how it is calculated.

Third, the Introduction and Discussion sections contain many repetitive statements and statements that lead the reader away from the main story line. The Introduction should focus on the effects of land-use intensity on soil organisms that here have been studied with DNA barcoding, offering a new level of resolution. Your focus and hypothesis on differences in responses between common and rare taxa is really secondary and less "a priori" than the main hypothesis that land-use intensity reduces soil biodiversity. The Discussion contains many speculations, in part related to this secondary hypothesis, that weaken the stronger results regarding the main hypothesis. Generally, we find it very difficult to say rare taxa can "better" indicate biodiversity responses to land-use intensity, because obviously you don't know what the "true biodiversity" is, which you implicitly use as reference.

Reviewer #1:

See above summary.

Reviewer #2:

In this paper the authors use meta barcoding to assess the diversity of soil invertebrates and relate diversity to land use. They find that more intensively managed habitats contain a lower diversity of soil fauna and, in particular, a lower diversity of rare (narrowly distributed) species. The diversity declines are consistent across groups: all groups decline or don't respond to land use, none increase. I think the paper is valuable in looking at soil fauna, which are poorly studied, and the results are interesting in showing that land use change has large effects on these taxa. It is also interesting that it is the rare species that decline with land use, as has been found in other organism groups. However, I have some reservations about the analysis and the framing of the study in terms of endemism.

1) The study aims to show the effect of land use change (conversion of native forests into grassland or perennial cropping) on soil fauna diversity. However, there is no attempt to correct for confounding factors. How were the sites chosen? Was there an attempt to ensure no confounding between land use and environmental factors such as altitude, soil type etc.? All that is said in the Materials and methods is that the sites were distributed across New Zealand. The Discussion mentions some possible confounding, as it is said that native forests are in more "rugged and less accessible areas". In addition, from Figure 6 it appears that many of the native forests are on the west coast of the South Island and there may therefore be climatic differences from the other sites? Currently there is no attempt to correct for confounding factors in the analysis: land use type is the only factor included in the models. I think it would be worth trying to fit some covariates in these models: climate variables and altitude would make sense and some soil variables such as soil type or pH would be good to consider too. This would make the analysis much more robust in showing that land use is the driver of soil fauna diversity, after correcting for soil and climate. It would also add interesting information on the other drivers of soil fauna diversity.

2) I found the term "endemism" confusing in the context of this study. There are no actual data on whether any of these taxa are endemic to a given area, as these data do not exist for soil fauna. Instead endemism is defined as the number of sites occupied by an OTU. Taxa occurring on few sites are therefore considered "endemic" but there is no information on whether they really are endemic or whether they occur elsewhere, e.g. outside New Zealand. Also, if a taxa is found in only two sites it would be considered endemic here but if those two sites were far apart, e.g. it was found in a northern and southern sample, it might not have a restricted distribution, it might simply be rare throughout the range. I therefore think that it would be clearer to talk about rare species and mention that here rarity is based on site occupancy. The situation may be even more problematic for phylogenetic endemism as in the original paper Rosauer et al., 2009, state that phylogenetic endemism values will be biased if closely related species occurring in other areas are missing from the sample (which is highly likely to be the case in this study).

2b) Related to this, I think it is important to mention in the Discussion that rarity is defined here relative to the sites sampled. Some of the species considered rare in this analysis might not be rare at all if they are common in habitats or areas not sampled in this study. There is nothing the authors can do about it this and I think the approach they have used is completely reasonable, but this limitation ought to be acknowledged in the Discussion.

3) How much is the measure of "endemic richness" affected by OTUs occurring in a single site? I could imagine that the distribution of these singleton taxa is quite stochastic and I wonder if it would be worth recalculating the index without them to check that they are not driving the whole pattern.

4) I am not really convinced that the analysis of R^2^ values (Figure 4) is valid. The R^2^ values for the different groups are not independent of each other as diversities of different taxa are likely to be correlated due to shared environmental responses or interactions between the groups. I am not sure that this analysis is really needed anyway, it is clear from Figure 3A that rare and phylogenetically distinct species respond more strongly to land use. However, if it is retained the authors should justify it and test the robustness of the analysis, perhaps with bootstrapping?

5) It would be useful to report the correlations between the diversities of different groups, in the supplementary information, to show which respond similarly.

6) The evidence that phylogenetic diversity responds more strongly than species richness is very weak. The R^2^ values and effect size of land use do not seem to differ between taxonomic and phylogenetic richness or endemism and phylogenetic endemism. The points in the fourth paragraph of the Discussion are therefore not supported by the analysis. If the authors want to check whether phylogenetic diversity does respond more strongly to land use then they should calculate measures of phylogenetic diversity uncorrelated with richness. This could be done in two ways: 1) by randomizing species between sites, whilst maintaining site richness values, and calculating expected phylogenetic diversity for the random communities. A standardized effect size would show if the observed phylogenetic diversity values are greater or lower than expected by chance (Webb et al., 2002). 2) the authors could extract residuals for phylogenetic diversity after correcting for taxonomic diversity and analyse the residuals. I imagine these approaches could also be used to correct phylogenetic endemism values. Also see Winter et al., 2013.

7) How well does the meta barcoding approach work to recover the species present in the sample? Are there data showing that it accurately recovers the species present? Did you check that species present in the Tullgren extracts were also recovered by the meta barcoding? It would be reassuring if there were some data on the robustness of the approach.

8) It is interesting that the authors find such a large effect of land use change on the soil fauna. Other studies in Europe have found that soil fauna are quite insensitive to land use intensification (Gossner et al., 2016). I imagine that this is because they looked at land use intensification within a habitat (i.e. within grasslands) while this study considers land conversion from forests to grassland and cropland. I think it would be worth commenting on this in the Discussion.

9) I feel it would be better to show plots of model predictions and standard errors, rather than the raw data boxplots currently used. If the models include covariates to correct for environmental confounders then this would allow the authors to plot the effect of land use after correcting for other factors.

10) Discussion, fifth paragraph: Collembola also seem resistant to land use change.

11) Table 1 should not be in the main text and could go to the supplement. If you want to use these post hoc tests then you could use letters in the figures to distinguish the land use types that differ from each other.

12) In Figure 3—source data 1B, it would be good to show effects (coefficients) from the models, rather than only showing the SS and F values.

Reviewer #3:

The paper by Dopheide and colleagues examines the effect of land use on soil faunal diversity. Overall, it is a nice analysis and examines a particularly under-studied community type, and I think it will be of value to the broader community. I do have some reservations about the presentation and to my mind, the most important concern of mine is #3 below.

1) There is quite a bit about homogenization in the inference in the Discussion, but this isn't actually tested for. You could do an analysis of β diversity measures (taxonomic and phylogenetic) to determine if communities are in fact more homogeneous in agricultural settings than in forests – see Swenson 2011 PloS ONE 6: e21264 and Jin et al. 2015, J. Ecol. 103: 742-749.

2) The metrics analyzed are good for comparing amongst one another, but I think another needs to be assessed. PD will be highly correlated with richness, and phylogenetic-endemism will pick up on phylogenetic distinctiveness as well as range size -two different possible responses to land use. I would recommend assessing MPD as well to see how land use influences the relatedness of species in communities independent of range size.

3) Phylogenetic-endemism needs to be clarified. How was endemism estimated? Was it across all 75 samples or within each land use type? Further, and this needs to be commented on in the manuscript, estimating endemism from 75 samples is dubious at best and better reflects habitat specialization than endemism per se. Forest specialists can be extremely widespread and not at all endemic. So, you need to be cautious about language and inference. I would recommend getting rid of 'endemism' altogether, except for the methods when describing the metrics, and instead refer to 'habitat specialists'.

4) Further, the comparison between individual species phylogenetic-endemism and community level aggregation can provide more insight as well. You could look at the distribution of species values and how these scale up to the community e.g., see Cadotte and Davies 2010 Div and Dist 16: 376-385 and R function in Pearse et al. 2015 Bioinformatics btv277. Though this latter suggestion is a recommendation and not a requirement since I have a vested interest, so feel free to ignore.

Detailed thoughts:

Abstract:

– There's no real scientific setup or overarching problem statement, just a statement that the authors want to compare something.

– It is not clear what 'community change' refers to. Externally driven or natural fluctuation or succession? I see Land-use later on, the first we see this is with the Results sentence. Needs a better set up. The first sentence of the Abstract should be about land-use driving diversity change and the need to adequately assess community sensitivity to land use.

Introduction:

Does a nice job of setting up the importance of the study, however, paragraph #1 is long and seems to make a number of different points. I prefer short and concise paragraphs that have single subjects and communication goals.

Results:

Subsection “Overall community composition”, last paragraph – it is not clear what heterogeneity refers to.

Figure 2: this is a difficult figure to distil meaningful information. Is there another way to show how lands alters abundance?

Figure 3B and C: can be moved to supplementary material if space is an issue.

Table 1: not needed, significant differences can be indicated on Figure 3A.

Overall, the results are strong and clearly support the major inferences.

Discussion:

Overall good and clear. Again 'heterogeneity' is unclear, please set this up better since it appears to be an important result. Further, there is little biology in the Discussion. Much of it is about the metrics and it would benefit from more discussion of the biology of the systems.

[Editors' note: further revisions were suggested prior to acceptance, as described below.]

Thank you for submitting your revised article "Rarity is a more reliable indicator of land-use impacts on soil invertebrate biodiversity than richness or diversity" for consideration by *eLife*. Your revision and response letter has been assessed by Bernhard Schmid as Reviewing Editor and Ian Baldwin as the Senior Editor.

The agreement was that we cannot yet proceed with the revision until you have more fully incorporated the major suggestions from the first round of reviews. You did provide extensive responses in your response letter, but did not add important items to you revision. These items are, again:

1) Make a linear contrast for the a priori hypothesis that is so pervasive throughout your paper, namely that there is a sequence in which you can put your five land-use categories (you even use this sequence in figures). There is nothing circular and statistically this is much more justified than multiple comparison. As mentioned before, making such a contrast is as valid as any other contrasts, and we can assure you that it will make a much stronger message. The way you can present it, is that you have a term with the five categories and four degrees of freedom and then the alternative of the linear contrast and the remainder with 1 and 3 degrees of freedom. The total of the SS of the two latter will equal the SS of the former but the MS for the linear contrast will be large. We don't mind if you only put this into the supplementary material, but we do want to see it.

Once your a-priori hypothesis that there is an effect of a continuous increase along the five categories, in spite of the inability to measure this continuity, has been tested highly significant, and the remainder, which tests for deviation from the a-priori hypothesis may even be far from significant, you will have a very strong message as you formulated it before but without statistical backing. To fit the linear contrast, just make an explanatory variable LUI with values 1, 2, 3, 4 and 5 for the land-use categories (LUC). Then fit LUI+LUC sequentially. Compare this with the fit of LUC without the linear contrast.

2) Add the analysis with rarest species excluded to the supplement.

3) We disagree with your statement "We think that this test overlooks important aspects of the results, namely the direction of biodiversity differences(as opposed to their consistency among taxa), and which taxa do or do not respond consistently. It is these aspects that the pairwise comparisons are intended to demonstrate." We believe the contrary is true. But it is your paper and we only would like to see the alternative analysis ((Metric ~ LUI+LUC+phylum+order+LUI:phylum+LUC:phylum+LUI:order+LUC:order) in the supplementary material. Note that the LUI-by-taxa interactions are exactly testing differences in direction of biodiversity responses to land-use categories!

4) Add an analysis with environmental covariates to the supplement.

Once we can see these items, we will progress with reviewing.

---

## [Author Response]

Essential revisions:The manuscript should be improved in three respects:First, there are several terms that are not well defined in the manuscript. "Land use" is sometimes considered as a state of an ecosystem and then called land-use category (please always use the hyphen when "land-use" is an adjectival noun) but sometimes considered as land-use change or intensification etc. Please use consistent terms throughout and make sure you define them well.

We have defined and reworded land-use terms for clarity and consistency throughout the manuscript, as suggested.

In this context and related to the second main point I suggest that you put the five land-use categories explicitly into a linear sequence from one to five and call it land-use intensity.

We thank the reviewers for this suggestion, which we have considered and explored in depth. Unfortunately, we encountered some fundamental issues that meant we were unable to implement it. This is partly because we cover forest, grassland and horticulture, which are all inherently managed very differently. This meant that we were unable to come up with a meaningful way of defining land use intensity that could apply across all land use categories. We tried ranking the importance of key management practices across the land uses (e.g. via pesticide and fertiliser use, disturbance frequency, vegetation complexity), but this failed to distinguish between plantation forestry and low-producing grassland, and between high producing grassland and perennial cropland. We considered using abiotic data as a way of quantifying intensity, but this effectively just tests whether that particular component of land-use intensity influenced diversity metrics. We also considered using the observed biotic data to infer land-use intensity rankings as suggested by the reviewers, but felt that this would create a new set of biases as it involves circular reasoning i.e. it effectively tests whether the gradient defined by biota influences biota. As we are unable to robustly define a land-use intensity gradient, we have more carefully worded the manuscript so that it is clearer that we are not looking directly at a defined intensity gradient.

Endemism is defined in the Materials and methods, but all reviewers would prefer a more generic term such as rare species because you cannot extrapolate from 75 sites to the entire area of your country, which would be necessary to know if a species that only occurred at 1 or 2 sites was geographically restricted.

We have replaced “endemism” with “rarity” (and “phylogenetic endemism” with “phylogenetic rarity”) throughout the paper.

You should also mention the caveat that some of your rare species could just be "sink" or "passenger" species not adapted to the site conditions, while your hypothesis assumes that they are habitat specialists.

We have added this point to the Discussion.

One reviewer suggests in this context that you could actually do a sensitivity analysis without the rarest species that only occur at one site.

We recalculated all biodiversity estimates with species that occur only in a single site excluded, as suggested. The resulting trends were largely consistent with those observed with all species included, although land-use differences were statistically somewhat weaker. We have noted this in the Materials and methods and Results. For example, mean overall rarity values per land use with OTUs that occur in just one site included were ranked natural forest > low-producing grassland > planted forest > high-producing grassland > perennial cropland, with significantly higher rarity in natural forest compared to all four other land uses (Figure 3). Mean overall rarity values per land use with these OTUs excluded were ranked in the same order, but the differences between land uses were less pronounced, with significantly higher mean rarity in natural forest compared to perennial cropland, but no significant differences between mean rarity values in natural forest compared to planted forest or grassland sites. Therefore, our results are partly, but not entirely, driven by species occurring in a single site.

Second, you should get much more from your data if you expand your analyses. In principle, you selected your five categories such that they can be ordered along a gradient. Best would be if you could calculate some value for land-use intensity for each category or even each site, but even in the absence of this you are allowed to simply assign land-use intensity values from 1-5 and use this linear contrast in all ANOVA (the easiest way is to code land-use categories from 1-5 and use it as continuous variable LUI and factor LUC: then fit LUI+LUC in this order and you get a stronger test of your main hypothesis, namely the linear contrast LUI, and a test for deviations from the linear contrast, which is measured by LUC when fitted after LUI).

We thank the reviewers for this suggestion. We have considered this point at great length. Unfortunately, as discussed above, we think this analysis approach is problematic. We are unaware of any applicable method for determining land-use intensity of our sites that applies across all land uses, and there is no clear a priori reason to rank all five land-use categories in a linear sequence. Any such ranking would be inherently circular (using our results to generate a ranking, then using that ranking as a predictor of our results). Therefore, we don’t feel we can carry out the ANOVA test as suggested.

Please replace all box-plots with means +/- standard errors.

We have replaced all box-plots with means +/- standard errors, as suggested.

In addition, you could show regression lines for the LUI-gradient wherever it is significant.

As we cannot determine a land-use intensity gradient nor order our land-use categories in a linear sequence, we are unable to carry out regression on this.

There would be the more sophisticated method of isotonic regression (Gaines, Steven D., and William R. Rice. 1990. Analysis of Biological Data When there are Ordered Expectations. The American Naturalist 135: 310-317.), but clearly the simpler method indicated above is also valid.The above will remove the need for the excessive use of pairwise comparisons which are statistically not independent and ill-suited to test ordered hypotheses. A further reduction of unnecessary repeats and more comprehensive analysis is to compare the responses of the different taxa to LUI(+LUC) in a single ANOVA by putting the metrics for the different taxa into a single column and add a column with the taxa identities (or even two, one for phyla and one for classes/orders). You can then fit the following ANOVA-model:Metric ~ LUI+LUC+phylum+order+LUI:phylum+LUC:phylum+LUI:order+LUC:order, or more simply without LUC terms and only using phylum or order. The interactions indicated by ":" test if the different taxonomic groups respond differently. Using the above you can exchange Figure 3—source data 1B and omit Supplementary Table 4.

Again, we thank the reviewers for this suggestion. As discussed above, we are unable to determine land-use intensity values for our sites or land-use categories, so cannot use the suggested ANOVA model. Given this constraint, we have instead added a variant of the suggested ANOVA (Metric ~ LUC * taxonomic group) to our analysis, to test for consistent responses of different taxa to land use.

We think that this test overlooks important aspects of the results, namely the direction of biodiversity differences (as opposed to their consistency among taxa), and which taxa do or do not respond consistently. It is these aspects that the pairwise comparisons are intended to demonstrate. The point about pairwise tests being ill-suited to testing ordered hypotheses is not applicable, because our land-use categories could not be ordered a priori, and we have been unable to identify a better statistical method for analysing the patterns of land use responses among different taxa. Consequently, we have retained the pairwise tests, but we have revised the Results section so that less emphasis is placed on these tests. This includes removing the tables of pairwise test results, and indicating significant differences directly on the figures, as suggested by reviewer 2 (below).

There is still the problem of comparing the R^2^ values (or better just use %SS, which are increments in multiple R^2^) mentioned by one reviewer and we ask you to consider their suggestions. It also seems you analyzed R^2^ values as (meta-)data; this requires explanation and justification in the Materials and methods, because R^2^ values will have a special distribution that deviates from normal.

We have revised and replaced the analysis of R^2^ values according to the suggestions above, and those of reviewer 2 (below), as follows. We used non-parametric bootstrapping stratified by taxonomic group to estimate % Sum of Squares explained by land use for each biodiversity metric and plotted these results as a histogram (Figure 5). We compared these values between metrics using a non-parametric statistical test. This has been explained in the Materials and methods.

Furthermore, as noted by one reviewer, from your Figure 6 it appears that the land-use categories are not well "randomized" across space and thus covariates could explain some of your results. You could include some of these first in the fitting sequence of ANOVAs, but we accept that you don't want to put in too many covariates because there are not so many sites to test more complicated models and sometimes correction for covariates can be unjustified if they are not a "true cause" of an observed effect.

The spatial randomisation of sites is constrained by existing patterns of land use. We agree that covariates could explain some of our results, but this is beyond the main intent of this analysis, which is to compare the performance of different biodiversity metrics for detecting impacts of land use on soil invertebrates. In practice, land use is a surrogate for many other variables (e.g. geology, vegetation, and soil chemistry), which may be the true drivers of invertebrate biodiversity metrics.

One reviewer also suggests that you should try phylogenetic metrics that are not mathematically related to richness.

We have added comparisons of mean pairwise distance, and standardised effect sizes for each of the phylogenetic metrics, between land-uses (Figures 3 and 4).

Your suggestion that the results imply homogenization needs to be tested statistically, which in fact can be done quite easily e.g. by calculating β diversities. You also use a term "heterogeneity", which obviously is taken from output of an R function. However, you cannot expect readers to check this up but rather need to give a clear definition of heterogeneity and how it is calculated.

We tested for multivariate homogeneity of sample dispersions between land-use categories using the “betadisper” function in the R package vegan. This test determines the dispersion of groups as the average distance of group members to the group centroid in multivariate space, and uses ANOVA to test for dispersion differences among groups. Our statements about homogenisation were based on this test. We have also added tests of β diversity and phylogenetic β diversity between land-use categories, as suggested, to further support our statements about homogenisation. We have clarified this in the Materials and methods and Results.

Third, the Introduction and Discussion sections contain many repetitive statements and statements that lead the reader away from the main story line. The Introduction should focus on the effects of land-use intensity on soil organisms that here have been studied with DNA barcoding, offering a new level of resolution. Your focus and hypothesis on differences in responses between common and rare taxa is really secondary and less "a priori" than the main hypothesis that land-use intensity reduces soil biodiversity. The Discussion contains many speculations, in part related to this secondary hypothesis, that weaken the stronger results regarding the main hypothesis.

We have revised the Introduction and Discussion according to these suggestions. For example, we have placed more emphasis on the impacts of land use in the Introduction and Discussion and have edited or removed various speculative and repetitive statements.

Generally, we find it very difficult to say rare taxa can "better" indicate biodiversity responses to land-use intensity, because obviously you don't know what the "true biodiversity" is, which you implicitly use as reference.

We agree that we don’t know what the “true biodiversity” is, and we have therefore replaced any statements about rarity being “better” with more precise terms. For example, the title now states that “rarity is a more reliable indicator” and the Abstract that “rarity provides more consistent and clearer evidence” of land-use impacts, rather than rarity is a “better” indicator.

Reviewer #1:See above summary.Reviewer #2:[…] 1) The study aims to show the effect of land use change (conversion of native forests into grassland or perennial cropping) on soil fauna diversity. However, there is no attempt to correct for confounding factors. How were the sites chosen? Was there an attempt to ensure no confounding between land use and environmental factors such as altitude, soil type etc.? All that is said in the Materials and methods is that the sites were distributed across New Zealand. The Discussion mentions some possible confounding, as it is said that native forests are in more "rugged and less accessible areas". In addition, from Figure 6 it appears that many of the native forests are on the west coast of the South Island and there may therefore be climatic differences from the other sites? Currently there is no attempt to correct for confounding factors in the analysis: land use type is the only factor included in the models. I think it would be worth trying to fit some covariates in these models: climate variables and altitude would make sense and some soil variables such as soil type or pH would be good to consider too. This would make the analysis much more robust in showing that land use is the driver of soil fauna diversity, after correcting for soil and climate. It would also add interesting information on the other drivers of soil fauna diversity.

As discussed above, the randomisation of sites is constrained by land use patterns throughout New Zealand. We agree that covariates would explain some of our results, but feel that this is beyond the intent of this paper, which is to compare the performance of different biodiversity metrics for detecting impacts of land use on soil invertebrates, rather than to understand the specific drivers of soil fauna diversity. We view land use in this context as a way of summarising a wide range of different drivers, including those described by the reviewer.

2) I found the term "endemism" confusing in the context of this study. There are no actual data on whether any of these taxa are endemic to a given area, as these data do not exist for soil fauna. Instead endemism is defined as the number of sites occupied by an OTU. Taxa occurring on few sites are therefore considered "endemic" but there is no information on whether they really are endemic or whether they occur elsewhere, e.g. outside New Zealand. Also, if a taxa is found in only two sites it would be considered endemic here but if those two sites were far apart, e.g. it was found in a northern and southern sample, it might not have a restricted distribution, it might simply be rare throughout the range. I therefore think that it would be clearer to talk about rare species and mention that here rarity is based on site occupancy. The situation may be even more problematic for phylogenetic endemism as in the original paper Rosauer et al., 2009, state that phylogenetic endemism values will be biased if closely related species occurring in other areas are missing from the sample (which is highly likely to be the case in this study).

These are good points, and we thank the reviewer for pointing them out. As mentioned above, we have replaced “endemism” with “rarity” throughout the manuscript.

2b) Related to this, I think it is important to mention in the Discussion that rarity is defined here relative to the sites sampled. Some of the species considered rare in this analysis might not be rare at all if they are common in habitats or areas not sampled in this study. There is nothing the authors can do about it this and I think the approach they have used is completely reasonable, but this limitation ought to be acknowledged in the Discussion.

We have acknowledged this point in the Discussion.

3) How much is the measure of "endemic richness" affected by OTUs occurring in a single site? I could imagine that the distribution of these singleton taxa is quite stochastic and I wonder if it would be worth recalculating the index without them to check that they are not driving the whole pattern.

As discussed above, we recalculated biodiversity estimates with these OTUs excluded, resulting in similar endemism richness (now called rarity) trends, but statistically weaker differences among land uses, indicating that these OTUs partially but not entirely drive the observed patterns. We have noted this in the paper (Results and Materials and methods).

4) I am not really convinced that the analysis of R^2^ values (Figure 4) is valid. The R^2^ values for the different groups are not independent of each other as diversities of different taxa are likely to be correlated due to shared environmental responses or interactions between the groups. I am not sure that this analysis is really needed anyway, it is clear from Figure 3A that rare and phylogenetically distinct species respond more strongly to land use. However, if it is retained the authors should justify it and test the robustness of the analysis, perhaps with bootstrapping?

As mentioned above, we have replaced the previous analysis of R^2^ values with a non-parametric bootstrapping analysis followed by a non-parametric statistical test, as suggested (subsections “Biodiversity differences among invertebrate taxa”, and “Biodiversity analyses and statistics”, Figure 5).

5) It would be useful to report the correlations between the diversities of different groups, in the supplementary information, to show which respond similarly.

We have added this information to the Appendix (Appendix—figures 7-12).

6) The evidence that phylogenetic diversity responds more strongly than species richness is very weak. The R^2^ values and effect size of land use do not seem to differ between taxonomic and phylogenetic richness or endemism and phylogenetic endemism. The points in the fourth paragraph of the Discussion are therefore not supported by the analysis. If the authors want to check whether phylogenetic diversity does respond more strongly to land use then they should calculate measures of phylogenetic diversity uncorrelated with richness. This could be done in two ways: 1) by randomizing species between sites, whilst maintaining site richness values, and calculating expected phylogenetic diversity for the random communities. A standardized effect size would show if the observed phylogenetic diversity values are greater or lower than expected by chance (Webb et al., 2002). 2) the authors could extract residuals for phylogenetic diversity after correcting for taxonomic diversity and analyse the residuals. I imagine these approaches could also be used to correct phylogenetic endemism values.Also see Winter et al., 2013.

We have calculated standardised effect sizes for each of the tested phylogenetic biodiversity metrics, as suggested (Figure 4, Figure 4—figure supplements 1 and 2). Phylogenetic diversity SES resulted in less consistent and different land use patterns among taxa compared to non-SES values. Phylogenetic endemism SES, in contrast, resulted in somewhat increased consistency of land use patterns among taxa compared to non-SES values, strongly suggesting that rarity declines from natural forest to agricultural land use. Together with the bootstrapping analysis described above, we think this does show that phylogenetic metrics do provide additional value.

7) How well does the meta barcoding approach work to recover the species present in the sample? Are there data showing that it accurately recovers the species present? Did you check that species present in the Tullgren extracts were also recovered by the meta barcoding? It would be reassuring if there were some data on the robustness of the approach.

Recovery of species using DNA metabarcoding is imperfect due to factors such as PCR primer mismatches causing PCR amplification biases to varying extents. However, previous work has shown that the approach tends to recover a majority of species in a sample, and enables the identification of many more species and a broader range of taxa than is typically feasible using traditional methods. We are therefore confident that our method provides a robust (albeit imperfect) assessment of soil invertebrate biodiversity.

8) It is interesting that the authors find such a large effect of land use change on the soil fauna. Other studies in Europe have found that soil fauna are quite insensitive to land use intensification (Gossner et al., 2016). I imagine that this is because they looked at land use intensification within a habitat (i.e. within grasslands) while this study considers land conversion from forests to grassland and cropland. I think it would be worth commenting on this in the Discussion.

This is an interesting point. We have added this to the Discussion, as suggested.

9) I feel it would be better to show plots of model predictions and standard errors, rather than the raw data boxplots currently used. If the models include covariates to correct for environmental confounders then this would allow the authors to plot the effect of land use after correcting for other factors.

The plots now show mean values and standard errors. As mentioned above, we think that including confounding factors is beyond the intention of this research.

10) Discussion, fifth paragraph: Collembola also seem resistant to land use change.

This is true, and we have updated the manuscript accordingly (Discussion).

11) Table 1 should not be in the main text and could go to the supplement. If you want to use these post hoc tests then you could use letters in the figures to distinguish the land use types that differ from each other.

We have removed Table 1, and added letters to figures to show differences between land uses, as suggested (Figures 3 and 4 and their figure supplements).

12) In Figure 3—source data 1B, it would be good to show effects (coefficients) from the models, rather than only showing the SS and F values.

The coefficients of a factorial ANOVA are simply the differences in the means compared to the first treatment level (intercept). We have added mean values to the figures, and therefore there is little additional information that would be gained by adding coefficients to the result tables. Rather, we think inclusion of coefficients in the results tables would make them overly complex.

Reviewer #3:The paper by Dopheide and colleagues examines the effect of land use on soil faunal diversity. Overall, it is a nice analysis and examines a particularly under-studied community type, and I think it will be of value to the broader community. I do have some reservations about the presentation and to my mind, the most important concern of mine is #3 below.1) There is quite a bit about homogenization in the inference in the Discussion, but this isn't actually tested for. You could do an analysis of β diversity measures (taxonomic and phylogenetic) to determine if communities are in fact more homogeneous in agricultural settings than in forests – see Swenson 2011 PloS ONE 6: e21264 and Jin et al. 2015, J. Ecol. 103: 742-749.

As discussed above, this was based on tests for heterogeneity/homogeneity of multivariate sample dispersions between land uses. We have now added tests of β diversity and phylogenetic β diversity among land uses. Together, these results provide evidence of biotic homogenisation.

2) The metrics analyzed are good for comparing amongst one another, but I think another needs to be assessed. PD will be highly correlated with richness, and phylogenetic-endemism will pick up on phylogenetic distinctiveness as well as range size -two different possible responses to land use. I would recommend assessing MPD as well to see how land use influences the relatedness of species in communities independent of range size.

We have added analyses of MPD to the manuscript, as suggested. In contrast to the other metrics, MPD does not differ among land uses for most taxa.

3) Phylogenetic-endemism needs to be clarified. How was endemism estimated? Was it across all 75 samples or within each land use type? Further, and this needs to be commented on in the manuscript, estimating endemism from 75 samples is dubious at best and better reflects habitat specialization than endemism per se. Forest specialists can be extremely widespread and not at all endemic. So, you need to be cautious about language and inference. I would recommend getting rid of 'endemism' altogether, except for the methods when describing the metrics, and instead refer to 'habitat specialists'.

Endemism and phylogenetic endemism were calculated across all 75 sites. We have clarified this in the Materials and methods. Furthermore, as discussed above, we have replaced “endemism” with “rarity” throughout the manuscript. We prefer to use “rarity” over “habitat specialisation” because the latter is only one of three axes of rarity and can’t be easily determined from our data.

4) Further, the comparison between individual species phylogenetic-endemism and community level aggregation can provide more insight as well. You could look at the distribution of species values and how these scale up to the community e.g., see Cadotte and Davies 2010 Div and Dist 16: 376-385 and R function in Pearse et al. 2015 Bioinformatics btv277. Though this latter suggestion is a recommendation and not a requirement since I have a vested interest, so feel free to ignore.

This is an interesting suggestion, but we think this may be getting beyond the aims of the manuscript and is likely to over-complicate it.

Detailed thoughts:Abstract:– There's no real scientific setup or overarching problem statement, just a statement that the authors want to compare something.

We have edited the Abstract such that it now begins with a statement of the problem.

– It is not clear what 'community change' refers to. Externally driven or natural fluctuation or succession? I see Land-use later on, the first we see this is with the results sentence. Needs a better set up. The first sentence of the Abstract should be about land-use driving diversity change and the need to adequately assess community sensitivity to land use.

We have revised the Abstract according to these suggestions.

Introduction:Does a nice job of setting up the importance of the study, however, paragraph #1 is long and seems to make a number of different points. I prefer short and concise paragraphs that have single subjects and communication goals.

We have revised paragraph 1 as suggested.

Results:Subsection “Overall community composition”, last paragraph – it is not clear what heterogeneity refers to.

As discussed above, heterogeneity referred to differences in multivariate community sample dispersion among land-use categories. We have clarified this in the manuscript, and added comparisons of β diversity to strengthen our results.

Figure 2: this is a difficult figure to distil meaningful information. Is there another way to show how lands alters abundance?

We respectfully disagree. The heatmap is intended to show the distribution of species across samples along with their relative abundances. There would surely be other ways to show changes in abundance in relation to land use, but we doubt they would be any more clear or concise, given the need to represent a large number of species.

Figure 3B and C: can be moved to supplementary material if space is an issue.

Figures 3B and C are now supplementary figures linked to Figure 3A (now simply Figure 3).

Table 1: not needed, significant differences can be indicated on Figure 3A.

As discussed above, Table 1 has been removed, and we have indicated significant differences on Figure 3, as suggested.

Overall, the results are strong and clearly support the major inferences.

Thanks!

Discussion:Overall good and clear. Again 'heterogeneity' is unclear, please set this up better since it appears to be an important result. Further, there is little biology in the Discussion. Much of it is about the metrics and it would benefit from more discussion of the biology of the systems.

We have clarified the meaning of heterogeneity throughout the paper, as suggested, and have added more discussion of land use impacts on soil communities.

[Editors' note: further revisions were suggested prior to acceptance, as described below.]

1) Make a linear contrast for the a-priori hypothesis that is so pervasive throughout your paper, namely that there is a sequence in which you can put your five land-use categories (you even use this sequence in figures). There is nothing circular and statistically this is much more justified than multiple comparison. As mentioned before, making such a contrast is as valid as any other contrasts, and we can assure you that it will make a much stronger message. The way you can present it, is that you have a term with the five categories and four degrees of freedom and then the alternative of the linear contrast and the remainder with 1 and 3 degrees of freedom. The total of the SS of the two latter will equal the SS of the former but the MS for the linear contrast will be large. We don't mind if you only put this into the supplementary material, but we do want to see it.Once your a priori hypothesis that there is an effect of a continuous increase along the five categories, in spite of the inability to measure this continuity, has been tested highly significant, and the remainder, which tests for deviation from the a-priori hypothesis may even be far from significant, you will have a very strong message as you formulated it before but without statistical backing. To fit the linear contrast, just make an explanatory variable LUI with values 1, 2, 3, 4 and 5 for the land-use categories (LUC). Then fit LUI+LUC sequentially. Compare this with the fit of LUC without the linear contrast.

As mentioned previously, we have some concerns with this approach. Our first concern is that after several years of research on land-use impacts on biodiversity, we have come to the realisation that the term “land-use intensity” is problematic. Trying to encompass all the axes of variation across diverse land-use categories into a single metric is probably impossible. Our second, related, concern is with the idea of deriving a ranking from the data and then using that ranking as a predictor of the data, which we view as circularity. Evidently the reviewers do not see this as a problem. Nonetheless, we feel that readers might misunderstand anything termed “land-use intensity” as an a priori, rather than derived variable.

We believe that these concerns can be managed through careful presentation and terminology. Rather than referring to “land-use intensity” we have used the term “derived land-use rank” or DLUR at relevant points in the manuscript. The advantage of this term is that it does not imply anything beyond how the term was quantified (i.e., it is simply a rank score, and it is derived). We have explained this approach in the Materials and methods section.

We carried out ANOVA tests for significant DLUR trends for each biodiversity metric and taxonomic group. This provided very similar conclusions to our tests with land use as a categorical factor. The results of these tests are found in Appendix (Appendix—tables 1 and 5).

2) Add the analysis with rarest species excluded to the supplement.

We have added the results of recalculated biodiversity estimates with the rarest species excluded (those that occur only in a single site) to the Appendix (Appendix—figures 3-5). These trends were largely consistent with those observed with all species included, although land-use differences were statistically somewhat weaker. For example, mean overall rarity values per land use were ranked in the same order with the rarest species included and excluded, but statistical differences between land uses were less pronounced in the latter case compared to the former.

3) We disagree with your statement "We think that this test overlooks important aspects of the results, namely the direction of biodiversity differences(as opposed to their consistency among taxa), and which taxa do or do not respond consistently. It is these aspects that the pairwise comparisons are intended to demonstrate." We believe the contrary is true. But it is your paper and we only would like to see the alternative analysis ((Metric ~ LUI+LUC+phylum+order+LUI:phylum+LUC:phylum+LUI:order+LUC:order) in the supplementary material. Note that the LUI-by-taxa interactions are exactly testing differences in direction of biodiversity responses to land-use categories!

We have added the suggested ANOVA to the paper, using DLUR instead of LUI. The ANOVA model used was as follows:

Metric ~ DLUR + LUC + group + DLUR:group + LUC:group

in which “group” is 17 taxonomic groups encompassing all the terrestrial invertebrate taxa detected.

This analysis has been explained in the Materials and methods section, with a summary of the outcomes in the Results, and full results in the Appendix (Appendix—table 2).

4) Add an analysis with environmental covariates to the supplement.

We have added analyses of environmental covariate effects on biodiversity to the paper. We did this in two ways. First, we carried out ANOVA tests for effects of spatial attributes (latitude and altitude) plus land-use category on overall biodiversity metrics. Second, we generated a PCA based on spatial and soil chemistry variables, then carried out ANOVA tests for effects of the major PCA components plus land-use category on overall biodiversity metrics. These analyses are described in the Materials and methods and Results, with the full details in the Appendix (Appendix—figure 6; Appendix—tables 3 and 4)